# Current Situation of Traditional Architecture Located inside Cultural Mayan Heritage Spaces in Remote Villages of Guatemala: Case of the Black Salt Kitchens

**Luis Pablo Yon Secaida [1],*, Suguru Mori [2] and Rie Nomura [2]**

[1]  Laboratory of Architectural Planning, Division of Architectural and Structural Design, N216, Engineering Faculty, Hokkaido University, Kita 13-Jo, Nishi 8-Chome, Sapporo 060-8628, Japan
[2]  Faculty of Engineering, Hokkaido University, Sapporo 060-8628, Japan; suguru-m@eng.hokudai.ac.jp (S.M.); nomurarie@eng.hokudai.ac.jp (R.N.)
*  Correspondence: secaidaluispablo.yon.m4@elms.hokudai.ac.jp

**Abstract:** In the town of Sacapulas, located in the mountainous country of Guatemala, there is a constant risk of natural disasters. Floods and landslides occur frequently, resulting in the loss of human lives and cultural aspects. Important to the region, the creation of the black salt is most affected. This resource has been created since the time of the Mayans on the salt beach surrounding the town. However, from the 1940s onwards, this industry has shrunk. As a result, architectural expressions known as "salt kitchens" have almost disappeared, and there is no information on the subject available. By employing interviews, area survey, and GPS mapping, it was discovered that the location of the salt kitchens is determined by the shape of the beach. However, only one third of the beach area is accessible up to this day. It was discovered that the destruction of the salt kitchens is due to natural elements as well as owners reusing the land for other economically viable functions. To preserve their existence, the first plans of the salt kitchens were created, and will help future researchers if necessary.

**Keywords:** cultural areas; space use; Mayan architecture

## 1. Introduction

This research is a continuation of the continuous effort to categorize, analyze, and recover the salt beach area of Sacapulas. Previous research has been conducted regarding the beach's current situation. In this paper, a part of the same original dataset will be used with new information gathered on site.

Culture and its inclusion in sustainable development has been researched lately, with more people being interested in the topic [1–3]. Sustainable development is defined as "development that meets the needs of the present without compromising the ability of future generations to meet their own needs" [4]. Its inclusion is based on three important aspects: social, environmental, and economical [1,2]. Examples can be given, such as the Sendai framework for disaster risk reduction, a joint effort seeking to reduce the impact of disasters over the next 15 years from its creation [5]. It is of the utmost importance to understand how an affected group of people reacts, adapts, and uses its resources to reduce the impact of disasters. Some groups, sadly, do not have the means to properly address such effects and seek external assistance to solve them [6]. This phenomenon has been researched lately, yet the efforts vary by country and region. This is true in European countries where specific care has been given. Countries in development still lack the interest and effort to reduce the loss of cultural heritage, especially the countries with a high degree of natural disasters [7].

A specific aspect of disaster management is related to culture heritage [7,8]. It is understood that climate change can affect culture, both the tangible [9] and intangible [10,11]

expressions that a group of people consider culture. A response to counteract this issue is the implementation of new approaches with a diverse set of tools, be it interdisciplinary, multidisciplinary, or transdisciplinary [12]. A four-pillar approach created by the US National Park Service (NPS) is a good example of this. The same could be said of UNESCO's Local and Indigenous Knowledge System (LINKS), which implements Intangible Cultural Heritage (ICH) into disaster management. ICH is knowledge shared by cultures all across the globe, recollected through ages of experience by local people. Another term for this knowledge is "Indigenous Knowledge systems and practices" (IKSP) [9,13]. The implementation of LINKS has been effective during the creation of new science and environmental policies [13].

Even though there has been an international effort towards reducing disasters, this has not been fully developed in the Latin American country of Guatemala. Research conducted in the country shows that there is an erosion of authority figures, resulting in slow decision making [14–16]. While not specifically using the LINKS, Guatemala does have disaster risk governance legislation, which is included in the legislative decree 109-96 called "Law on the National Coordinator for Disaster Reduction" [17]. This law was based on the Yokohama Strategy and Framework for Action, which gives priority attention to developing countries [18]. Currently, due to the lack of enforcement of said law, local disasters are normally addressed with foreign cooperation such as USAID [19]. Yet, the international help is not enough to address less-known communities such as the town of Sacapulas. This town is located in the mountain range of the Chucumatanes, Guatemala. To the north of the village is the Chixoy River, which flows to the south of Mexico. The importance of this town is found in the production of black salt, a traditional product with more than a thousand years of tradition. The creation of black salt is directly linked to three main actors: the people directly involved in the industry, known as salt makers; the architecture created specifically for the industry, known as "salt kitchens"; and the space where they collect the materials for the salt, called "Black Salt Beach", located north of the village, next to the Chixoy River. Historically, the Maya society has always been vulnerable against natural disasters [20–22], but there has been an increase in recent years [23]. Proof of this has been the constant flooding of the Chixoy River in the area. As a consequence, the black salt industry and every aspect linked to it (tangible and intangible) have been affected. Of the countless people that were part of this industry previously, only four people in an advanced stage of life remain [24].

Considering that Guatemala is a country with a high risk of natural disasters, and a high lack of adaptive capabilities [25], swift action must be taken as soon as possible. In fact, research to safeguard the black salt culture has been few and far between, focusing only on the process of black salt creation. Interest in the subject has been diverse, and mainly by foreigners. One of the most important research projects in the area was performed by Anthony P. Andrews in 1983 [26], where he described how the black salt created in Sacapulas has always been of interest both to people outside of the culture (Spanish conquerors and researchers), and people inside the region, such as the nearby kingdom of Quiche, which was interested in the town and its resources. Overall, the town of Sacapulas and the black salt have been an important product of the Guatemalan culture and a point of interest to many people across the years. The previous known research was conducted in 2002 and it focused specifically on the black salt, until 2023, with an effort to understand the current situation of the beach [24,27]. This research focused on understanding how the situation evolved since the 2002 research conducted in the area. It was found that there is a constant lack of information regarding the physical aspects of the space, and also the cultural expressions that were born due to the salt industry. Without a concrete point of comparison, it was decided to instead develop the first spatial information in the form of planimetry, photography of the current state, and activity division of the area based on the accounts of people interviewed. The analysis identified that, compared to the accounts of previous research, the area in fact shrunk and was overrun with more soil than was reported before. This has affected the space and the salt industry in ways that have yet to

be researched, such as the architectural elements used for salt cooking. There is also a lack of research focused on the relation between the space, its features, and how the architecture functions. This is because research in the area is usually performed by people not interested in these topics, and also because all of the research does not come from the viewpoint of an architect. This has become an important topic as there is the threat of a resilient culture, and cultural expressions that are valuable due to their historical context, being erased from memory. Understanding how to properly adapt a local LINKS system, it must first identify what is at stake to be lost. At this moment, we know what the current situation is, yet we need to understand how the past impacted and transformed the salt industry and its specific architecture.

## 1.1. Case Area

Guatemala is a country located in Central America, represented in Figures 1 and 2. It has borders with Mexico to the north, Belize and the Caribbean Sea to the northeast, Honduras to the east, El Salvador to the southeast, and the Pacific Ocean to the south. The country's territory is primarily mountainous due to the "Sierra de Los Cuchumatanes," a mountain range throughout Central America. As a result, topographic elevation varies from 500 m to 3800 m. Rural towns and cities are located all across this terrain variation, resulting in many people living in mountainous, remote areas.

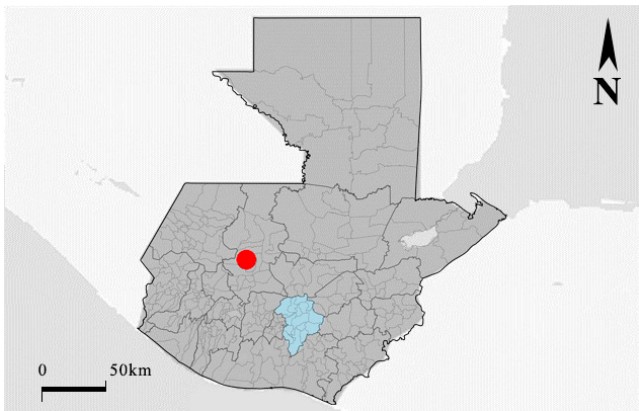

**Figure 1.** Geographical location of Guatemala. Research area, in red. Capital of Guatemala in blue.

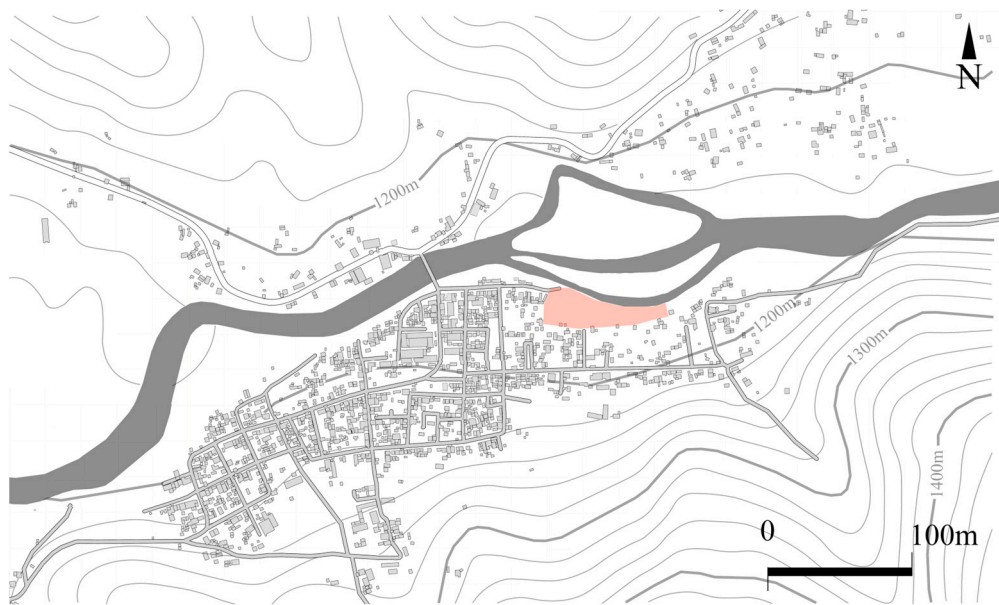

**Figure 2.** Sacapulas Town in the south of the Chixoy River, with the salt beach in red.

Inside the Cuchumatanes mountains is located the town of Sacapulas. While the region has developed steadily, communication is slow due to its current location, resulting in difficulties sharing the local culture. These difficulties have resulted in almost no current research being conducted in the area. Next to the town is the Chixoy River. This river runs through much of the northwestern territory of Guatemala and flows into southern Mexico. Across the river can be found salt deposits in the form of white spots, while the beach presents a source of thermal water. Both of these resources are used in the process of the black salt, which Figures 3 and 4 depicts. Because of its traditional way of being made, when cooked, it results in a black color with attributed medicinal properties that the local population benefits from [26,27].

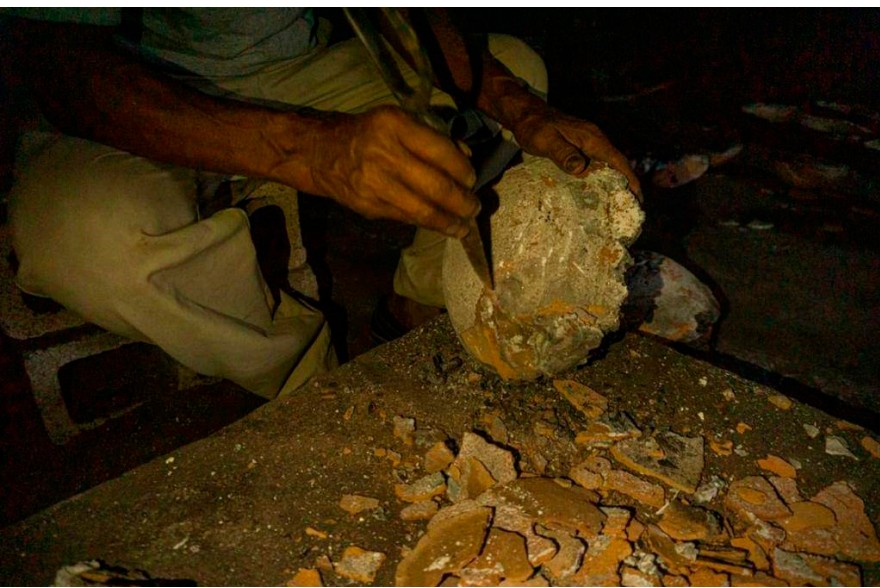

**Figure 3.** Black salt production; the mold is being broken to take the salt out. RAICES NGO (2023).

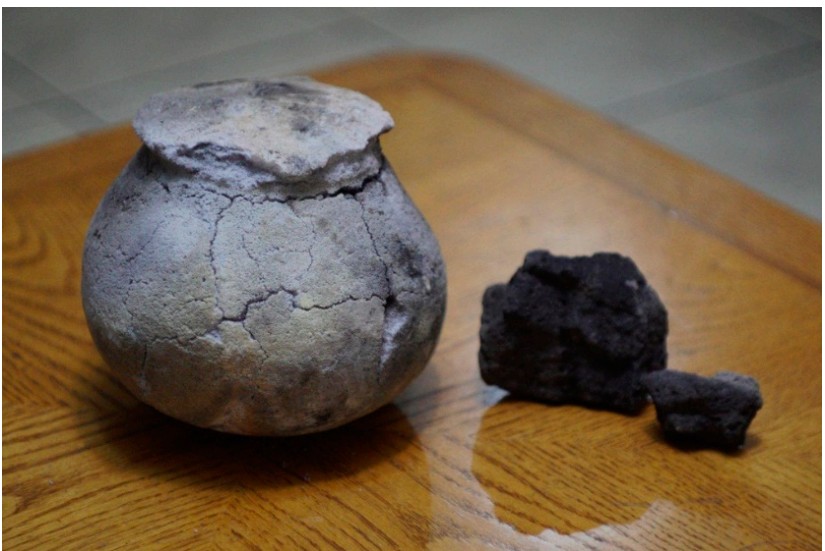

**Figure 4.** Black salt product. Author (2022).

Current research in the area has successfully identified the situation of the salt beach. At this moment, the salt beach comprises a surface area of 4873.42 m$^2$, of which almost 57% is covered by soil. As mentioned before, this soil comes from constant flooding from the Chixoy River. Of this area, only 5% has been excavated and is used for the salt industry.

The rest of the area is used for what seems a small array of activities, such as using the small shore of the beach to bathe due to hot water found in the area; cattle rising occurs all across the covered salt beach; and people use the space to move within the town of Sacapulas.

### 1.2. Objective

At this moment, research conducted towards the spatial and architectural features of the area has yet to be conducted. The importance of such a point of view resides in the relation between these two subjects and how they adapted due to the natural disasters of recent years. Therefore, this research is aimed specifically towards the black salt kitchens. It seeks to identify their current situation, usage, location, and relation to the black salt beach. Secondarily, it seeks to document the architectural information of the kitchens through architecture planimetry.

### 1.3. Limits

Considering the amount of information yet to be researched about the subject, several limits were imposed on this research. Firstly, this research will not consider the other cultural expressions that exist in relation to the black salt other than the architectural elements. To locate the kitchens, only people related to the black salt industry are considered, as they present the information firsthand. Finally, the research area is located across the current black salt beach. Previous research suggests that salt kitchens were located at the north of the black salt beach, yet this area is currently flooded and access to it is impossible.

### 1.4. Current Literature
#### 1.4.1. The Salt in Guatemala

The nature of this research is not about the overall situation and history of the Salinas, but a general understanding of its characteristics will be discussed to understand its context, from an architecture perspective. The traditional salt industry in Guatemala spans the entire Pacific Coast from the Suchiate River on the Mexican border to the border of El Salvador. According to Andrews, several hundred salt sources are located along the shores of the Pacific. According to Andrews, all of the salt produced on the Pacific coast of Guatemala was made by a process known as "cooking salt", which has more than 2000 years of history. Figure 5 depicts the major salt sources according to Andrews, which are located mostly on the west of Guatemala. There have been accounts by Spaniard chroniclers that described in detail the process to cook the salt. It is generally understood that it involves a three-step process: placing the soil impregnated with salt, filtering the soil with estuary water, and then cooking the result inside a clay or ceramic pan ("*olla*" in Spanish). While Andrews mentioned replacement of the pans with a more convenient vessel made from iron called "*perol*", this eventually transitioned to the usage of ovens. Interestingly enough, Sacapulas uses what is known as a "salt kitchen" as a space where the salt is cooked. Plenty of modern research mentions the salt kitchens as a novelty and an important factor for the fame of the Sacapulas salt. Yet, a specific analysis of this architectural element has been extremely scarce.

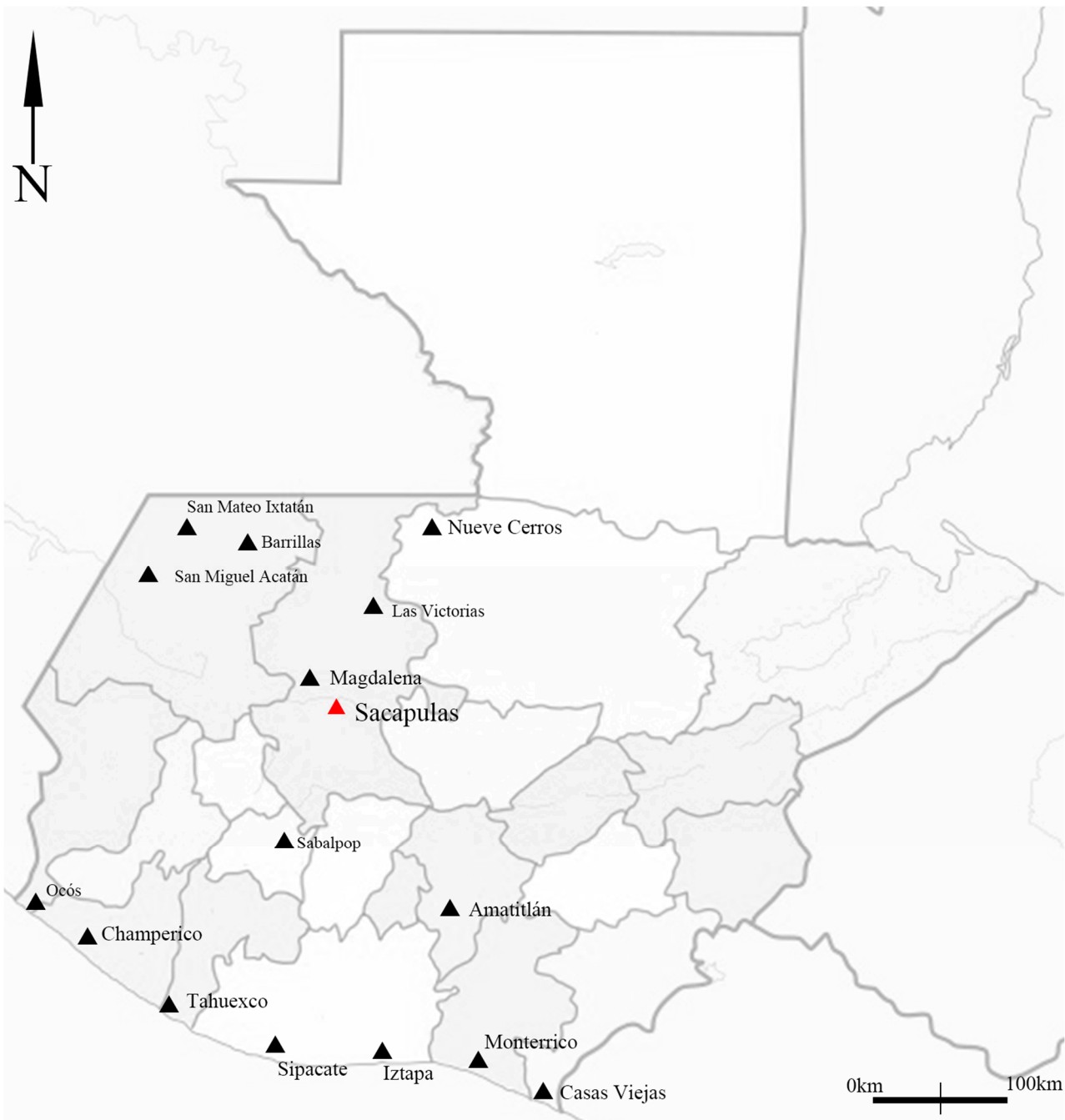

**Figure 5.** Historical salt sources identified by Andrews in his salt research. In red, Sacapulas. Andrews [26].

1.4.2. Salt Industry Formation

Thanks to archeological data found in the region of Ocos and Champerico, it is understood that the industry of salt began around the Formative Mayan period and was underway 2000 [26] years ago. Meanwhile, Spaniard chroniclers mention the existence of the salt industry to a degree of novelty [26,28], being conducted exclusively by natives. While the territories used for the salt industry were eventually divided with Spanish colonizers, the salt industry remained stable until the introduction of new technological innovations in the region. The previously mentioned ceramic pans, iron vessels, and ovens did increase its production, but the introduction of solar technology marked a noticeable shift in the local production. In fact, Andrews mentioned the increased use of

solar evaporation as the main reason for the decline of cooked salt in the region. This is true, as, comparatively, solar evaporation yields more product with less effort [26]. The usage of this technology was slow though; it was introduced during the 1920s and it became the dominant method of salt creation until the 1950s.

There are mentions of the complications due to the salt making process, both inside and outside of the salt industry, by Andrews, influencing its disappearance in the future. Firstly, the lack of natural resources such as firewood results in its disappearance. Secondly, the benefit of large agricultural and ranch spaces has led to deforestation. It is important to note that rural areas in Guatemala benefit mostly from these two economical activities, which makes them more attractive to people. In turn, cooked salt is both labor-intense and low yield compared to other salt methods.

### 1.4.3. Salt Making and Sacapulas Importance

The value of Sacapulas salt goes beyond its traditional method of production, as cooked salt is not specific to the town. Instead, its value comes from the historical background that it represents to Mayan history, the cultural expressions born from said industry, and the attributed value that is given by people that know about this product. Specifically, Sacapulas black salt is considered a famous type of salt due to its attributed health benefits, though no research has been conducted to validate or deny such a claim. Figure 6 depicts the process in which it attains its famous black color: the salt cooking process. Nevertheless, it has been reported by many researchers as an important factor of its notoriety [26,27]. This kind of salt making specifically uses an area known as "Salinas", which comprises a beach located north of the town of Sacapulas, and south of the Chixoy River. Most of this space is currently covered with alluvial deposits due to the constant floods caused by the river, which both Andrews and current research have confirmed as a constant issue. Specifically, Andrews mentions that at the time of his research, the space was covered with "as much as 2 m of alluvial deposits". Historical accounts place Sacapulas in a favorable position due to its location in relation to major trading routes. Its location is along the Chixoy River, which constitutes the east–west salt trade in the region. It is also located in a favorable position towards the Atlantic lowlands due to the Chixoy River being connected to the Usumacinta River. Its location was attractive to the Quiche region, as its location made it an important economical center. This resulted in the kingdom of Quiche eventually incorporating the town into its control until the arrival of Spanish conquerors. Colonial and modern history has shown considerable information on the Sacapulas industry, with recounts such as Titulo de los senores de Sacapulas and Spanish chronicles as important resources to understand the position that the town had for the Spanish crown and local Mayan people. Current research has focused on the actual state of the beach, and understanding how it has evolved due to the decline of salt making in the region. Traces of the importance of Sacapulas salt making still exist. An example of this is the current trading routes in the region. According to Andrews, the trading routes used by Mayas during the precolonial times were adapted by Spaniards and even used by current traders. Such examples exist, like the trading routes towards south and central Quiche, which are a pattern that dated to the XIV century when Sacapulas was integrated into Quiche. It was also mentioned that the salt route to the west was very old. This is due to the fact that salt tributes were found in the western region of Guatemala, tributes paid with Sacapulas salt [26].

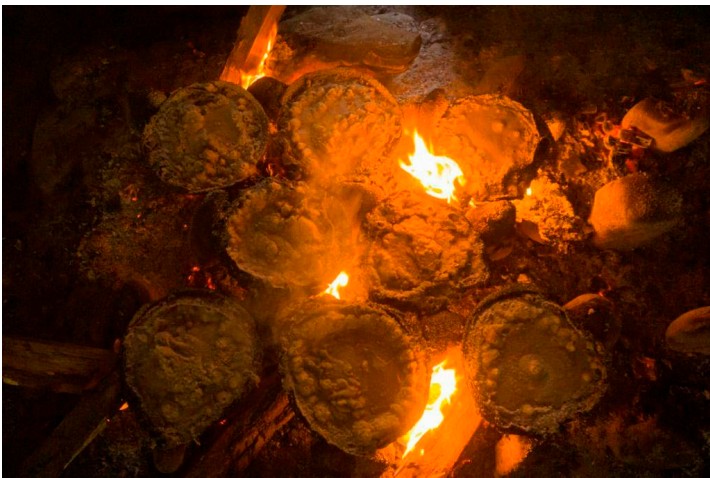

**Figure 6.** Black salt cooking process. RAICES NGO (2023).

Historically, the importance of the black salt industry is found in several mentions by Spanish conquistadors and Mayan mythology related to the region. Specifically, chronicles written after the conquest mention the Spanish conquistadors' interest in salt creation. The earliest mention of salt is a report by three Franciscans in 1579 [28]. This report relates the conquistadors' need for salt and how, for this reason, salt had to be obtained from Sacapulas. The salt makers were referred as "salineros" (Salt workers), and received a part of the salt as payment, which they eventually sold in their communities. The next mention of salt consisted of impressions by the Spanish colonizers. This recollection was written and reported by Martin Alfonso Tovilla, the mayor of the province of Vera Paz, Golfo Dulce, Sacapulas, and Manchen. His report to the Spanish crown was compiled in the book "Historicas Dyscriptivas" (Descriptive stories) in 1629. On this occasion, he mentions the importance of salt in the gastronomy of the locals, as they "do not do or want anything more than corn tortillas and a little chili and salt, which makes them happier than if they were eating turkeys…" [29].

Tovilla also mentions that the area's importance was also perceived as high because it was one of the few regions which produced salt. This importance was critical with the political situation of Sacapulas, as the riches were a point of issue with the Quiche kings and their kingdom located in the ruins of "Gumarkaaj", south of Sacapulas (around one hour of driving). Prisoners of war were commonly captured and forced to work in the salt industry. Tovilla concludes by giving his opinion regarding the low price of black salt compared to the volume of wood necessary for its creation, finding the process too exhaustive for the price it is sold. He said the rate was around fifteen "cakes" (salt molds) for one "*real*" (currency at the time).

Little research has been conducted in recent years, the latest being on the importance of black salt. This is found in the anthropological book "Traditions of Guatemala," which defines black salt as "a resource of great importance" [27]. In addition, there is evidence that various tributes were made in the form of salt during colonial times, proving that salt was considered an essential and valuable product for the Spaniards and the natives [26].

1.4.4. Salt Kitchens—Architecture Specific to the Industry

The area known as the black salt beach has been afflicted by constant flooding. This is due to the location of the Chixoy River in relation to the town of Sacapulas [24]. The result of the constant flooding and loss of space is a reduction in people active in the black salt industry. Consequently, the tools used in salt making have also ceased to be used. In this case, the "traditional salt stove" is one of the essential tools in salt making, known locally as "salt kitchens". Figure 7 is an example of this architecture. These constructions are concentrated above the black salt beach and comprise two essential elements: the main kitchen and the distillation box or dripping area. The kitchen comprises an enclosed space

needed for cooking salt. It is a small room excavated underground, surrounded by stone walls [30]. While the construction of the walls is rudimentary (it does not present any internal structure), the gable roof presents a wood structure.

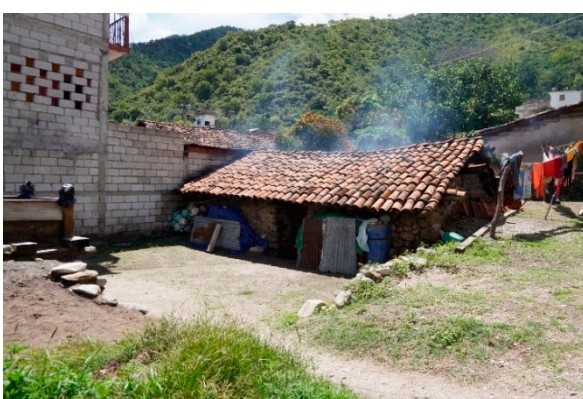

**Figure 7.** Salt kitchen. Author (2022).

1.4.5. Cajón and Pila de Destilación (Distillation Box and Basin)

In front of the kitchen, on the outside, is the salt preparation area represented by Figure 8. It comprises the distillation box denominated "cajon", an element usually 1.68 m in diameter and 1 meter deep. While the main structure is of wood, the box floor is made of wooden rods or cane and matting, named "petate". The box is placed on top of stone walls, on a mound of earth, at a certain height. Underneath, the distillation area or basin is built. Its function is to distill the salt using the thermal waters found around the beach shore. This step is crucial as it happens before cooking the salt in the kitchen. The basin is made of brick and stone, glued with mortar. The basin has the necessary capacity for eight "tinajas": a traditional plastic jar used in rural areas.

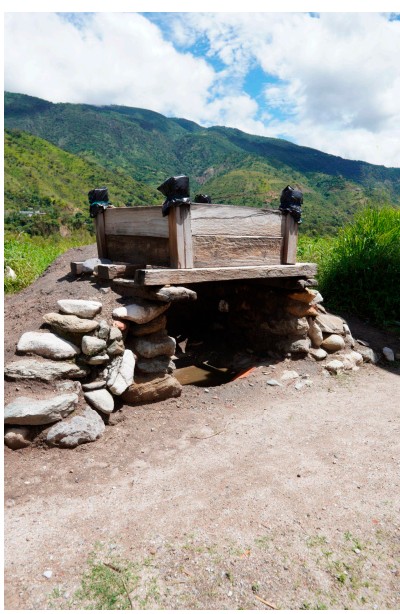

**Figure 8.** Distillation box and area. Author (2022).

1.4.6. Features of the Salt Kitchen

The architectural language found in the kitchens is of something other than Maya origin, which suggests that they were created after the pre-hispanic period. This fact is alluded to by Tovilla's text, which does not mention the use of specific architecture to create salt. Also, the use of clay tiles in roofs as an architectural element started after the

conquest, and did not originate from the region. However, Ruud Van Akkereen mentions in "Titulo de los señores de Sacapulas" that the pre-hispanic name of Sacapulas comes from the name "Tuja" or "Tujalá" [31]. He mentions the possibility that tuja refers to the low houses where people worked the black salt. Thus, there is a possibility that this architecture existed in pre-hispanic times in a different shape. There are discrepancies regarding the height of the salt kitchen. According to Reyna, the total height of a kitchen is 2 to 2.5 m (one meter underground, the rest over ground level), with measurements of 6 × 8 m. Ordoñez mentions between 3 and 4 m in height, with no mention of its measurements. The primary tool found inside the salt kitchen is the "heart of the fire," a slight mound of earth measuring 1.67 × 5.02 × 0.41–0.84 m wide, located in the center. On it, 40 tenamastes are placed: pointed stones taken from the Chixoy River. Over the tenamastes, "cajetes" or pots made from animal feces or clay are placed with the previously distilled water inside. These are usually used only once due to the fragility of the material. Also, the differences in temperature result in many of these pots being broken after they are used (Figure 3). The salt creation process is detailed in Section 1.4.7.

New kitchens are not built due to the small size of the salt industry. For this reason, maintenance must be performed to keep them usable. Maintenance focuses specifically on the roof structure comprising wood. This is replaced every two or three years due to the high intensity of the fire. In contrast, the tiles used on the roof are cleaned, not replaced. Informants estimate that the tiles are over 100 years old [27]. Due to the scarcity of kitchens, Ordoñez reports that salt kitchens can be borrowed or rented. The price is Q10 (USD 1.27) each time they are used or Q100 (USD 12.77) for a season from January to April. Another important fact is that the "heart of the fire" can be shared among salineros, which can help divide the cost of rent between two or more workers. Finally, throughout the off-season, salt makers aim to extract enough soil with salt to cook throughout the year. Thus, the salt kitchen also serves as a warehouse storing potential economical revenue to the salt maker.

### 1.4.7. Salt Creation Process

Understanding the value of the black salt's architecture and industry requires knowledge of how the product is made. The process initially mentioned by Alfonso Tovilla is the same as the one used in 1978, reported by Reyna and Monaghan, and by Amilcar Ordoñez in his 2001 research [27,30]. It should be noted that the process Tovilla mentions is described in general terms compared to Reyna and Monaghan's work. An element that has not changed has been the tools used in the salt creation process, which implies that the work process has not been optimized with new technologies. The process mentioned by Tovilla, and Reyna and Monaghan consists of two main parts: the "sowing of the salt"; and the cooking and subsequent finalization of the product. The first step of the process ("sowing the salt") is a method of salt germination involving the beach's specific temperature and soil obtained from it. Due to the nature of this research, the process is summarized in the most important steps for its creation. Therefore, this is a compilation of the steps mentioned by both Reyna and Monaghan and Ordoñez in their respective investigations.

As mentioned previously, salt production only occurs during the dry season of the year (November to June). However, the salted soil obtained during these months is stored and produced as salt throughout the year. Usually, the salt kitchens serve as a salt creation area and salt soil storage for future work. The work begins in the morning when the salt maker spreads special, clean soil on his or her part of the beach, no more than one centimeter thick. The documentation does not mention why the soil used is special. The land of the beach is measured by "petaquilla," a specific informal measurement unit that equals 12 square varas (Spanish measure equal to 8.385 sqr m). Then, he fills a tinaja with thermal water obtained from one of the many open wells on the beach. With half of a gourd, he hydrates the soil so that later the sun heats the beach enough to make the salt sprout and impregnate the soil. According to the report, this process is conducted for three consecutive days until the salt maker, from his own experience, decides that the soil has enough salt impregnated.

The next step is the mobilization of the soil to the salt kitchens. Figures 9 and 10 depicts this part of the process.

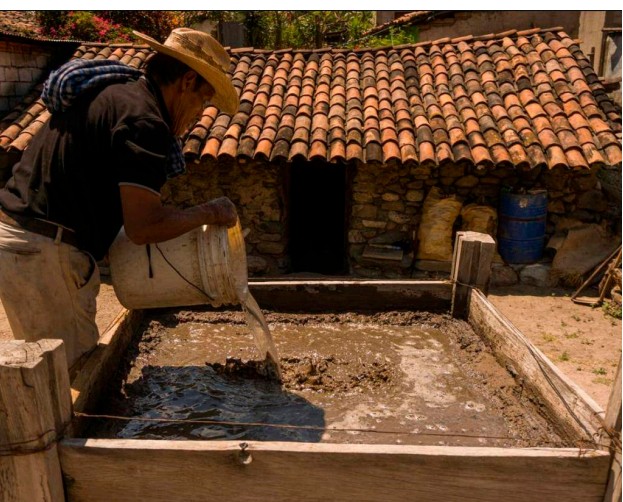

**Figure 9.** Distillation process. Hot water from the springs is being poured over the soil impregnated with salt. RAICES NGO (2023).

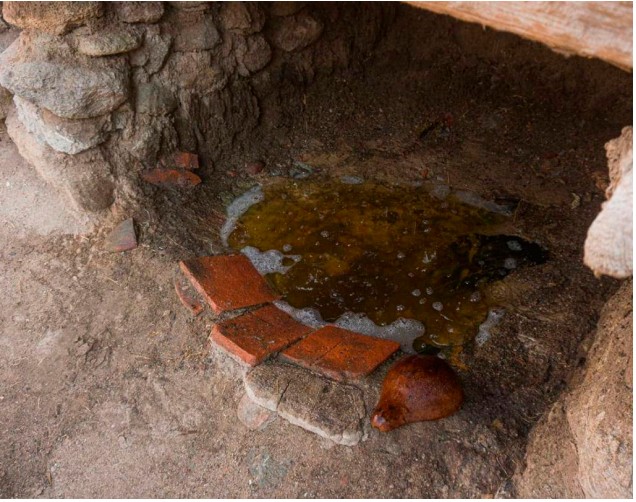

**Figure 10.** Lower part of the distillation box. RAICES NGO (2023).

The salt is transported to the salt kitchens on foot. Here, the soil will be distilled one petaquilla at a time. The soil is transported to the front of the kitchen, where the distillation box and basin are located. The salt is placed in the box, separated by a few m from the ground. The next day, the hot water from wells on the beach is poured in, a process that takes from 5 a.m. to 3 p.m. (10 h). At the end of the process, the soil is removed, and more is hauled back to start again. The result is distilled water with salt, or "aguasal". With the filtered salt water, the salt maker creates the "heart of fire", a rudimentary oven in which the salt molds are placed on the 40 tenamastes. The cajetes are heated with salted water inside and cooked for 5 to 6 h to evaporate the water. This process is called "burning" the salt. The next day, the pot is broken, and the salt lid is removed, ready to be sold. Figures 11 and 12 depicts this process.

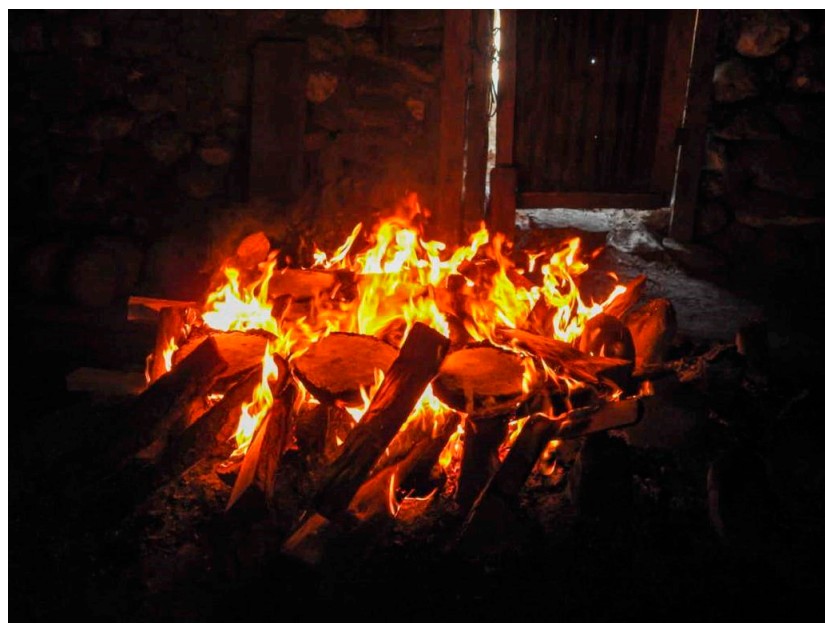

**Figure 11.** The cooking space known as "The heart of fire". RAICES NGO (2023).

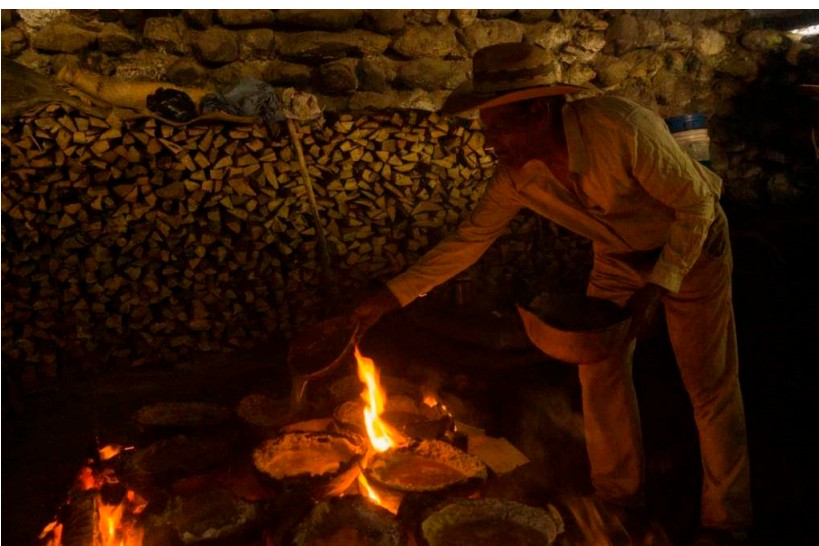

**Figure 12.** Cooking process. Source: RAICES NGO (2023).

The resulting product of the salt workers is twofold: Atzam and Xuupej. Atzam is small pieces of white salt resulting from the burning of the aguasal three times. However, reintroducing the atzam into the heart of fire results in Xuupej, black salt [27]. At the time of Reyna and Monaghan's work, approximately 250 pounds of salt were produced in two days and sold at high or low prices, depending on the season. This fact does not seem to have changed according to Ordoñez's research, where he mentions that each time salt is made, between 1 and 2 quintals (150 to 300 pounds) are produced. Figures 13 and 14 depict the last hours before the salt is finished.

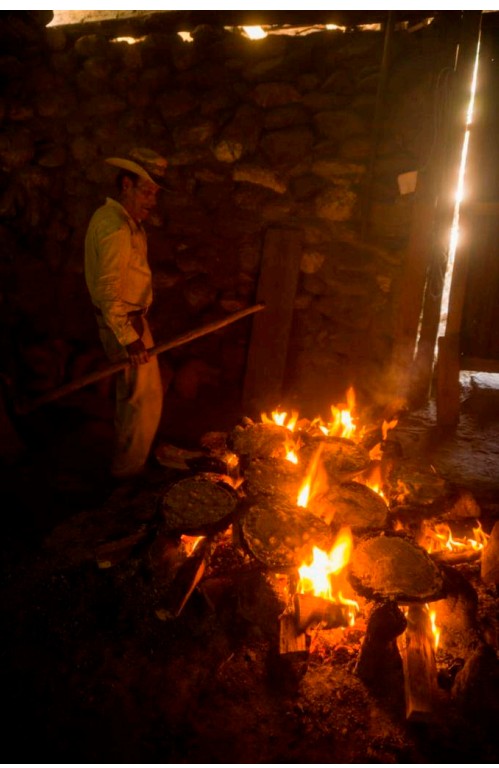

**Figure 13.** Final hours until the cooking is finished. RAICES NGO (2023).

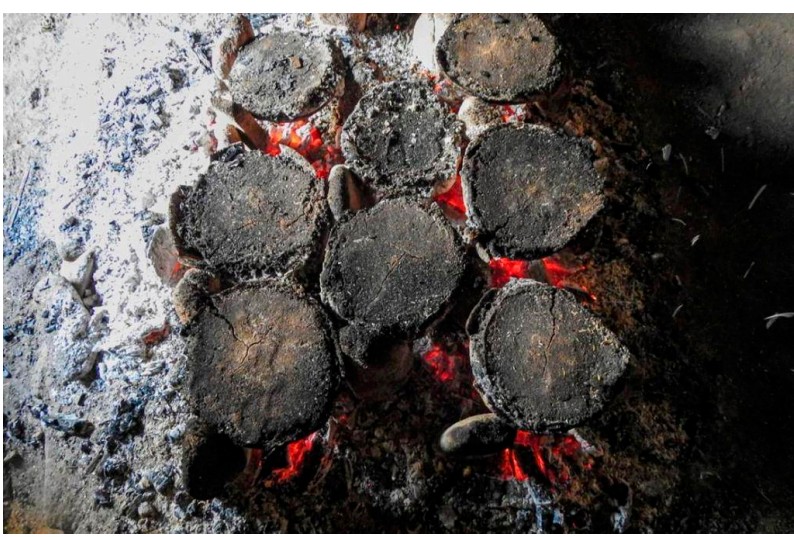

**Figure 14.** Finished black salt. RAICES NGO (2023).

1.4.8. Number of Salt Kitchens

The total number of kitchens is a mystery; in 1981, Reyna and Monaghan mentioned the existence of 11 salt kitchens. Meanwhile, Andrews mentioned on his 1983 research that there were 14 salt kitchens standing. However, in 2002, according to Ordoñez, only 6 kitchens were still in use during his research. According to Ordoñez's informants, there were around 16 kitchens in operation previously. Ordoñez did not define the specific year the 16 kitchens were in use. It can be hypothesized that the small number is due to two factors: a reduction in the black salt industry results in the reinterpretation of the space of the salt kitchens for another, more economically attractive use. And second, the historical earthquake of February 1976, where most of the buildings suffered damage [32]. According to Ordoñez, many areas of the black salt beach are now used as corrals and pigpens.

## 2. Materials and Methods

Black salt is a research topic wide enough but yet to be researched to a degree that access to information on it would be considered easy. Therefore, the lack of information has resulted in a complicated starting point for any person interested in the subject. A good way to address this challenge is by implementing not only one research method, but a diverse array of techniques that cover a broader set of research questions. This is known as "triangulation" [33]. The idea of triangulation is obtaining data from difference sources, varying the methodology, researcher, and/or theories. The triangulation approach is not an end in itself, but instead a tool to better understand a complex problem while also being effective in the time employed during the research. It also serves as a way to reduce the amount of personal bias the researcher might have, as different sources of information yield a more complete result. Finally, the relationship between quantitative and qualitative methods benefits from a triangulation approach, as it gives a better reach to the research while confirming or denying information from interviews. In the case of this research, the triangulation approach was used to complete the resulting dataset, with different information groups used for this process. Specifically, using the Denzin approach to triangulation techniques, "Methodological triangulation" was used. Of this group, "between method" triangulation was selected. This triangulation approach focuses on the usage of a variety of methods to collect and interpret data, and is used where two or more distinct methods are employed to research the same phenomenon—in this case, the relation of the salt kitchen and the salt beach. There is evidence of this approach being helpful in understanding disasters and local issues in Guatemala [14]. A summary of the analyses performed is shown in Table 1, while Figure 15 depicts the methodology employed and the workflow conducted.

**Table 1.** Types of analysis performed in this research.

| Method | Data Sources | Time | Space | Researcher | Goal |
|---|---|---|---|---|---|
| Interviews | People related to black salt (former workers, traders, and current salt workers). | Interviews were conducted twice; the first was conducted over one week as the pilot survey. The second one was conducted in person for one week, with a visual analysis of the space. | Town of Sacapulas | A principal investigator together with a 2-person working group in the area. | Information about the historical use of the area and current situation. |
| Salt kitchen measurements | On-site measurements of one salt kitchen and its features | Carried out in the same timeframe as the primary survey | Salt kitchen location | A principal investigator, together with an assistant | General planimetry of the area and analysis |
| Salt kitchen's location | On-site visit, GPS localization | Carried out in the same timeframe as the primary survey | Beach area surroundings | A principal investigator, together with an assistant | Location of the salt kitchens and spaces where they were built. Documentation of how the architecture was distributed concerning salt beach. |

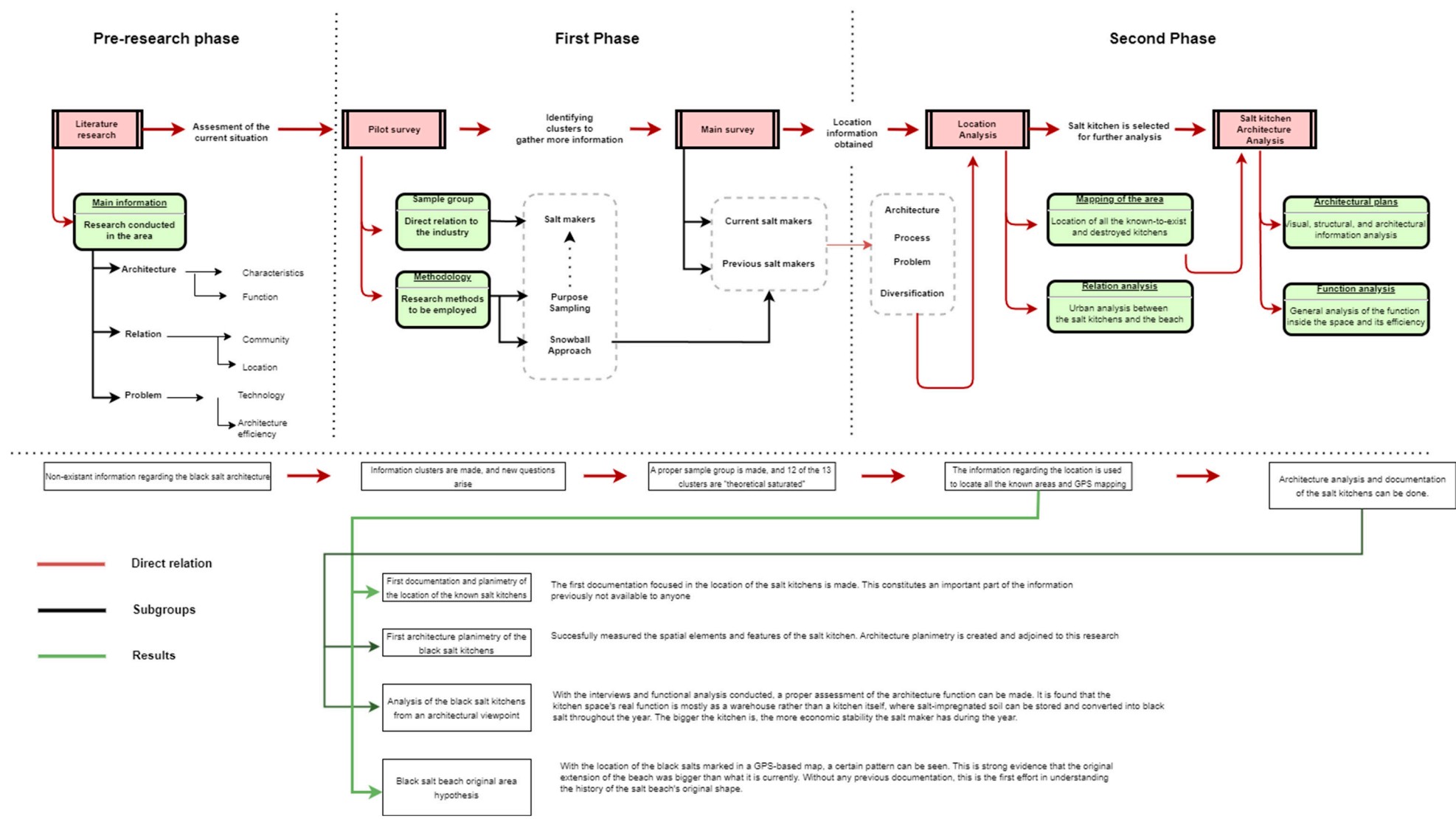

**Figure 15.** Methodology diagram.

After selecting the triangulation method, we need to define the types of analysis to be conducted in this research. Three analysis types were selected for this research, ranging from qualitive to quantitative. The main analysis and information were obtained by interview. A certain part of the information obtained was used originally in the previous research conducted in the area [24], yet part of the unused information was utilized on this follow up research by implementing new information specifically from the salt kitchen. This analysis was conducted in 2 parts, the first being an approximation of the area conducted in March to April 2022 as a pilot survey, and the second one being conducted in September 2022 as the main survey. The pilot survey was focused on specific interviews of a sample group of people who had a relationship with the black salt, using the "modified ground theory" as the main methodology employed. The "Modified Ground Theory" approach is based on the systematic method developed by Glaser and Strauss called "Ground Theory", in their 1967 book [34]. This qualitative method requires that the researcher start the investigation with pure data and no available theoretical background or knowledge of the phenomena. This is quite a challenge in the case of the black salt due to the complexity of the situation and the lack of information available: it is required to know the value of the industry and its relation to the people of Sacapulas. Therefore, a method which implements rough definitions and theories previous to the start of the research is needed while using the general "ground theory" methodology. This is called the "Modified Ground Theory", and it facilitates the researcher by taking into account the context of the black salt beach and its kitchens [35]. The result is a main dataset comprising qualitative information in the form of interviews, which was further completed in the second phase. The second phase was completed in September 2022 and focused on deepening the information obtained in the pilot survey by conducting a main survey. A further description of this method is found in Section 2.2.1.

The next analyses are considered secondary due to their being dependent on the information retrieved by the interview phase. These are the "location analysis of the salt kitchens" and "Salt Kitchen measurements". Location analysis is based on validating the information given by the interviewees regarding their knowledge of the salt kitchens. Due to the inexistence of these data, it required asking and then physically going to the spaces that were mentioned. After this, an inventory of their situation and usage of the terrain was conducted to update the current information of the architectural elements. This was supported by the usage of GPS tools to "map" their locations and eventually compare the data to the physical urban thread and distance between them. This helped not only understand the position of each element, but also to identify if there exists a pattern or a relation with the salt beach.

"Salt Kitchen measurements" refers to the measurement and construction analysis conducted on one of the remaining salt kitchens found in the area. Due to not knowing the specific location of any of the remaining salt kitchens (or their existence), this part of the research also depended on the previous two analyses conducted. This analysis focused on determining the general measurements of all the structural and architectural elements that comprise the salt kitchen. While information regarding these elements roughly exists in previous research [30], it is not concise and does not present the required architectural documents to be used by people interested in the salt kitchens. In contrast, this analysis seeks to provide the measurements and visual information of each element found in the space while also analyzing its function. This is further explained in Sections 2.2.2, 2.2.3 and 3.1.

After defining the analysis to be conducted, the methodology was developed as a step-by-step process, dividing the overall research into two steps: the first analysis phase and the second analysis phase. The first analysis phase consists of the literature research and the interview stage. The literature research comprises the research of all available data on the subject on the basis of previous research in the area. All this has been discussed in Sections 1 and 2. Meanwhile, the interview analysis comprises a pilot interview and a subsequent main interview. The resulting information was catalogued and analyzed, which

helped move to the second phase. The secondary phase comprises the location analysis and salt kitchen analysis. This is further explained in Sections 2.1.2, 2.2.2 and 2.2.3.

## 2.1. Materials Descriptions

### 2.1.1. Interview

As previously mentioned, the same sampling group that was part of the previous research will be used in this one. Yet, a different dataset extracted from the result will be used in this part of the research with additional data obtained from the secondary analyses conducted in the space. The original sampling conducted in this study is based on 17 people with a relationship with the black salt industry. They are divided into those who have maintained or had a direct relationship. This group is divided by gender, resulting in seven men and ten women. Due to the initial lack of information on the subject, it was decided to use the snowball approach, which helped increase the group of correspondents in the main interview. The snowball approach comprises asking a starting sample group for more people with knowledge of the subject of the research. This "snowballs" into more and more people while retaining the same level of relationship to the research question [36]. This results in two surveys (one conducted in the first phase, or the "pilot survey", and a second one in the second phase, or the "main survey"), with a noticeable difference in size. The pilot survey resulted in four correspondents. Meanwhile, the main survey correspondents increased in number up to 11, shown in Table 2. Figure 16 shows the total division of gender between both surveys conducted. The name for the subjects is the "Main survey subject" (MSS).

**Table 2.** Main survey sample subjects.

| No. | Ref. Name | Age | Relationship with Salt | Current Job |
|---|---|---|---|---|
| 1 | MSS-A | 75 years | It is carried out | Currently in the industry, diversifies his income by farming in the wet season. |
| 2 | MSS-B | 72–75 years | It is carried out | Diversifies the income with local candy named "alpinique" |
| 3 | MSS-C | 75 years | It is carried out | - |
| 4 | MSS-D | 46 years old | Father performed | Bakes cakes and bread thanks to the oven his father built. |
| 5 | MSS-E | 49 years old | Father performed | Makes ice cream and works with her husband in metal plating and jewelry. |
| 6 | MSS-F | 71 years old | Mother made | No information |
| 7 | MSS-G | 74 years old | Worked on it | Lives with her husband. She did not share the information |
| 8 | MSS-H | 48 years old | Worked on it | No information |
| 9 | MSS-I | 50–55 years | Worked on it | Works in a taco shop near the central plaza. |
| 10 | MSS-J | 57 years old | Worked on it | Along with other products, they are small merchants/resellers. |
| 11 | MSS-K | 84 years old | It is marketed | Along with other products, they are small traders/retailers. |

**Table 2.** *Cont.*

| No. | Ref. Name | Age | Relationship with Salt | Current Job |
|---|---|---|---|---|
| 12 | MSS-L | 78 years old | The marketer/family realized | Along with other products, they are small traders/retailers. |
| 13 | MSS-M | 85 years old | It was marketed | Switched to sugar candy because it is easier to sell. |
| 14 | MSS-N | 71 years old | I grew up with her | Not shared |

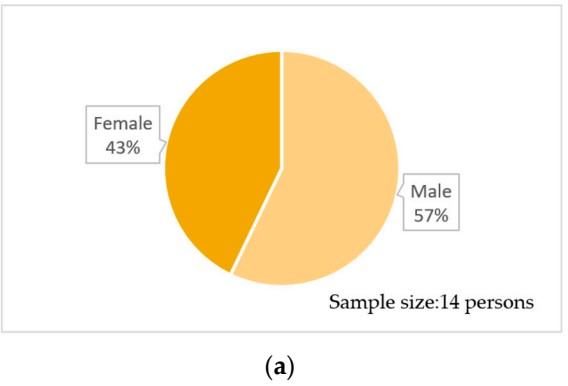

(**a**)

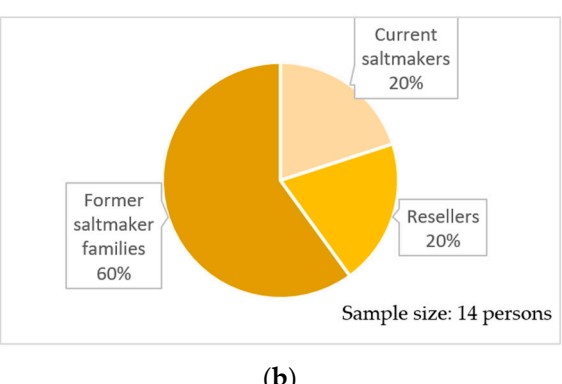

(**b**)

**Figure 16.** (**a**) Pie diagram of the total percentage of respondents on the second survey. (**b**) Pie diagram of the relationship of the respondents with the black salt industry on the second survey.

### 2.1.2. Salt Kitchen Measurements and Area Location

Analyzing all the data obtained from past research, it was concluded that information related to salt kitchens and their architecture has been extremely scarce. The only data found by Monaghan and Reina are relatively general and function as a description of the area rather than an architectural analysis. It was decided to obtain, catalog, and digitize the architectural information of the salt kitchens, inside and outside of them. The process involved the researcher, his assistant, and the owner of the salt kitchen. Obtaining the information comprised mostly a site visit to the MSS-A salt kitchen, where the measurements of the external and internal elements were identified. It was initially expected to analyze the three kitchens that are still being used. Yet, due to the privacy of rural Mayan communities and the researcher being an outsider, only one of the salt makers (MSS-A) agreed to this. Though according to MSS-A, the construction and space elements are the same in every one of the kitchens. This analysis was conducted with a research assistant, and took place during the wet season; thus, there was no inconvenience in disturbing the daily salt cooking of MSS-A.

After understanding the general aspects of the architecture, the research was expanded to the surrounding areas to find their whereabouts. Based on the information previously obtained in the interviews, the information collected in both the interviews was used as a basis to find the ruins of the salt kitchens. A subsequent field survey of the area near the salt beach was conducted. The objective was to understand the terrain and find if there was any relation between their distribution on the salt beach, between each salt kitchen, and the current usage of the spaces where salt kitchens were built.

### 2.2. Methodology

#### 2.2.1. Interview

The first pilot survey was conducted over one month due to the location and availability of the people involved. The goal of this survey was to understand the actual situation of the beach, how many people are still working in the black salt industry, and what kind

of information can be obtained from the interviewees. As mentioned before, a snowball approach was initially used to help achieve these goals, which helped further deepen the understanding of the black salt beach in the main survey. In total, it resulted in 4 h of conversation. There were several challenges in gathering information due to the area's location and respondents' age. To overcome this issue, questions were reinterpreted in the second survey and a temporary stay was conducted in the town. The questionnaire used is presented in Appendix B.

During the pilot survey, the information obtained was analyzed line by line using microanalysis and separated using the methodology employed by Kambaru [37]. For this purpose, all the answers obtained in the interviews were transcribed verbatim. Then, each of the responses was analyzed to find similar patterns. This resulted in "information groups" that present the same meanings that relate to similar concepts which are depicted in Table 3. These groups will be referred as "information clusters" and the aim of this method is to reach "theoretical saturation". This means that once enough information is obtained from any of these clusters, the concept is considered to have reached "theoretical saturation" and no more information is needed to be gathered. The "Theoretical saturation" is a limit that has to be decided by the researcher, though. In the case of the pilot survey, there is no prior knowledge of which information clusters will be made. The only information gathered was from the literature research available at the moment, and even then, it has been mentioned that the amount of information on black salt is still scarce on topics outside the salt industry. Therefore, saturation was considered when the information gathered was similar to the previous literature research. As such, only the "Process" information cluster was too well researched that new information was null. In total, 10 information clusters were identified, ranging from the relation of the black salt industry with the interviewee, to the influence of black salt beyond the town of Sacapulas (such as clusters "Religion" and "Trade"). An example of the microanalysis tool can be found in Appendix C.

**Table 3.** Pre-survey information clusters.

| No. | Concept | Definition | No. of Iterations Pre-Survey | Theoretical Saturation? |
|---|---|---|---|---|
| 1 | Relation | What is the relationship between salt and the people of Sacapulas? | 11 | NO |
| 2 | Product | Any product related to black salt (such as black salt, salt water, and hot springs.) | 5 | NO |
| 3 | Future | What is the future you envision for black salt? | 4 | NO |
| 4 | Usage | The use of black salt as a product | 7 | YES |
| 5 | Religion | Religion-related aspects performed | 8 | NO |
| 6 | Architecture | Everything related to architecture related to production, or people working in black salt. | 1 | NO |
| 7 | Terrain | Related information on the spatial area used for the creation of the black salt | 3 | NO |
| 8 | Authorities | Information about the authorities' perspective | 4 | NO |
| 9 | Process | Information related to the process of creation of black salt | Theoretical saturation | YES |
| 10 | Trade | Price information and sales locations | 1 | NO |

During the main survey, the results were analyzed in the same way as in the pilot survey. In total, the main dataset comprised 171 new entries divided into 13 information clusters, and 3 of the new information clusters were identified by recurring mentions in the interviews conducted. Their importance was also noted to the relevancy of the investigation effort conducted in the area. Thanks to the pilot survey, it was understood what information was necessary to deepen and obtain about the different clusters previously identified. For this, the main survey consisted of pinpoint questions that sought to deepen the understanding and successfully reach a logical conclusion or perspective. This is

the "theoretical saturation" that was considered. After the microanalysis conducted on the information, 12 of the 13 reached "theoretical saturation". The only one that could not reach this point was "authorities", which is a cluster focused on the intervention of local authorities to manage the salt beach. At the point of developing this research, communication with the authorities has been unsuccessful. The summary of the new information is found in Table 4. Due to the nature and objectives of this research, only the relevant information clusters will be used in the analysis and discussion. As mentioned before, the nature of this research paper focuses on a specific topic matter; thus, only a part of the dataset is relevant and will be employed. The first research paper made about the subject used "Terrain", "Usage", "Religion", and "Ownership" clusters in the research process. This leaves 9 information clusters that could be triangulated with new information about different aspects of the subject matter. In this paper, it was decided to use the information recapitulated in the "Architecture," "Process", "Problem," and "Diversification" clusters.

**Table 4.** Main survey information clusters.

| No. | Concept | Definition | No. of Iterations Pre-Survey | No. of Iterations Primary Survey | Theoretical Saturation? |
|---|---|---|---|---|---|
| 1 | Relation | What is the relationship between salt and the people of Sacapulas? | 11 | 31 | YES |
| 2 | Product | Any product related to the black salt process (such as white salt, black salt, salt water, and thermal water). | 6 | 12 | YES, extended with triangulation |
| 3 | Future | What is the future you envision for black salt? | 4 | 6 | YES, interviewees mentioned the same desire |
| 4 | Usage | The use of black salt as a product | Theoretical saturation | - | YES |
| 5 | Religion | Religion-related aspects performed | 8 | 20 | YES |
| 6 | Architecture | Everything related to architecture related to production, or people working in black salt. | 1 | 12 | YES, extended with triangulation |
| 7 | Terrain | Related information on the spatial area used for the creation of black salt | 3 | 32 | YES, extended with triangulation |
| 8 | Authorities | Information about the authorities' perspective | 4 | 1 | Authorities could not be contacted |
| 9 | Process | Information related to the process of creation of black salt | Theoretical saturation | - | YES |
| 10 | Trade | Price information and sales locations | 1 | 17 | YES |
| 11 | Problem | Any problems encountered when making black salt | 0 | 15 | YES |
| 12 | Ownership | Current land ownership by salt producers or outsiders | 0 | 20 | YES, extended with triangulation |
| 13 | Diversification | Changes in economic income due to salt | 0 | 5 | YES |

The objective of this research paper is to fully understand the situation of the salt kitchens and its relation to the salt beach. Therefore, the cluster "Architecture" was selected due to it containing all the relevant information about the salt kitchens not previously mentioned by other research. At this point, the information gathered on this cluster mainly comprises the known location of the active salt beaches and the known location of previous salt kitchens (currently destroyed salt kitchens). "Process" was also selected due to the analysis of the overall function of the kitchen. This helped understand how the architecture works towards the end goal of salt cooking, and a further analysis was made with the secondary analysis conducted. "Problem" and "Diversification" are clusters that present both the issues that the salt industry is facing, and what the response by salt maker families

to this phenomenon is. Specially, how it affected the cultural expressions that this research is interested on. All of this information was eventually triangulated with the information from the secondary analysis performed in the area, giving a bigger and more complete outlook and analysis.

### 2.2.2. Architecture Measurements and Planimetry

In this research, we chose to analyze in detail the measurement of the black salt kitchens. This type of work has yet to be conducted in detail before, the mention of it by Reyna and Monaghan being the only available data on the subject. This has resulted in a lack of critical information of certain aspects about the process: What is the main tool used for salt cooking, what are its dimensions, and what is its overall function? Specifically, the information obtained in this analysis comes from the viewpoint of an architect. Therefore, the information obtained corresponds to the following: construction elements used, architectural elements, and spatial measurements inside and outside the architectural elements. After that, the data were entered into the Revit 2023 architectural modeling program, where the architecture's views, plans, sections, and general details were created. The usage of this program was due to its convenience and easiness to complete the task at hand, and because it presents the results not only in 2D (planimetry), but also in 3D (visualization of the overall building). The results can be found in Appendix A with a detailed analysis of its features and spatial use in Sections 3.1 and 3.1.1.

### 2.2.3. On-Site Architecture Localization

Analyzing the location of each salt kitchen, two essential things could be determined. First, there is no record of the locations of any salt kitchens; only the knowledge of the former salt makers provides a general idea of their location. Second, a record of the salt kitchens currently in use is needed. Since this information has not yet been developed, it is not easy to understand the current state of the black salt industry. By obtaining the location of salt kitchens and the spaces where they existed, it is possible to understand their relationship to the salt beach, which would help define the shape of the beach before the floods buried the majority of it. Two sets of information were used at this stage:

- The collective memory of the people related to black salt. This has been obtained in the interviews and analyzed previously. The results were several areas in which salt kitchens had been built but are now destroyed.
- Field analysis of the spaces to determine if there are residues of previous constructions. The excavation of a rectangular space determines the construction of a salt kitchen. This excavation is not backfilled and can be easily identified unless the space is reused for growing crops. An example of such a space is found in Figure 17.

The result of the field visit was the location of fifteen areas where salt kitchens were built and three existing ones in use. Due to privacy concerns, some of the space owners did not share information other than the location of the salt kitchen. Meanwhile, other locations were found run over with vegetation. A summary of these data is presented in Table 5. By knowing the location, it was then pinned in a GPS and mapped in Google Maps, resulting in a semblance of an orderly distribution across the area. With these results, it can be much better understood if there is a relationship between the space, the kitchens, and the salt beach. The locations of all the elements found is represented in Figure 18 while a discussion is conducted in Section 3.2.

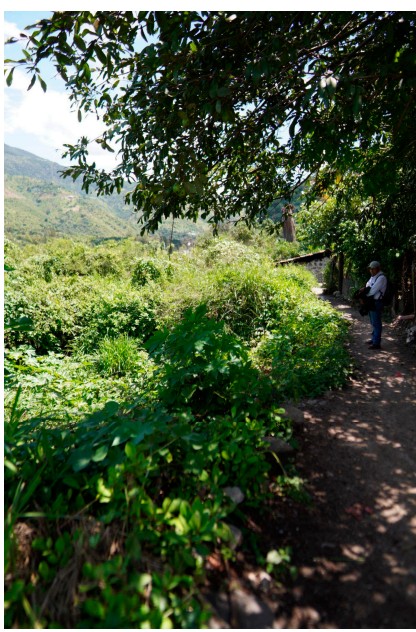

**Figure 17.** Area where a salt kitchen was built, currently destroyed. Author (2022).

**Table 5.** Summary of salt kitchens' situation and distance between each of them.

| No. | Situation | Current Situation | Distance between the Closest Salt Kitchen |
|---|---|---|---|
| 1 | Destroyed | Idle | Point 0 |
| 2 | Destroyed | Idle | 58.72 m (1–2) |
| 3 | Destroyed | Planting | 20.03 m (2–3) |
| 4 | Destroyed | Idle | 11.98 m (3–4) |
| 5 | Destroyed | Planting | 48.44 m (4–5) |
| 6 | Existing | - | 14.54 m (5–6) |
| 7 | Destroyed | Construction | 26.40 m (6–7) |
| 8 | Destroyed | Construction | 26.75 m (7–8) |
| 9 | Existing | - | 30.25 m (8–9) |
| 10 | Destroyed | Idle | 10.03 m (9–10) |
| 11 | Destroyed | Idle | 9.98 m (10–11) |
| 12 | Existing | - | 10.10 mts (11–12) |
| 13 | Destroyed | Idle, private land | 22.14 mts (12–13) |
| 14 | Destroyed | Idle, private land | 13.75 m (13–14) |
| 15 | Destroyed | Cattle breeding (?) | 49.75 m (14–15) |
| 16 | Destroyed | Idle | 41.86 m (15–16) |
| 17 | Destroyed | Idle | 17.27 mts (16–17) |
| 18 | Destroyed | Idle | 8.43 m (17–18) |
| 19 | Destroyed | Idle | 8.72 m (18–19) |

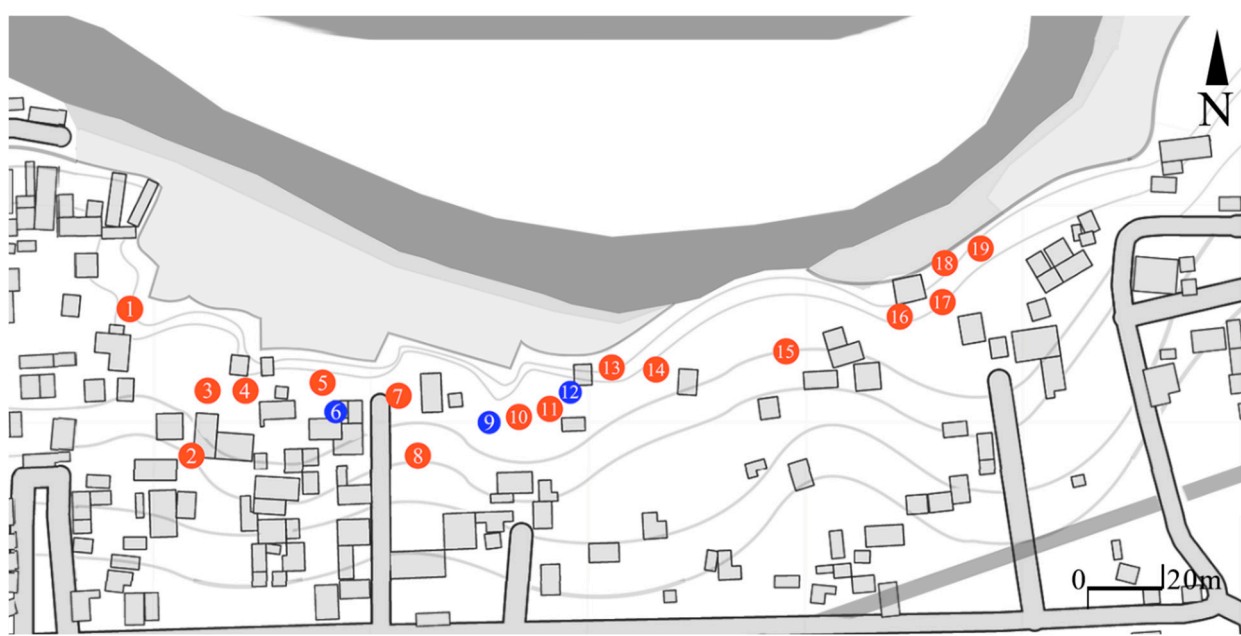

**Figure 18.** Locations of the salt kitchens. In blue, the existing ones, in red, the empty spaces of previous kitchens. The numbers depict the salt kitchens mentioned in Table 5.

There is some difference in the number of salt kitchens found compared to the information obtained in the existing literature and the interviews conducted with the people involved. According to MSS-A, the number of salt kitchens in 1970 was around 44. However, it was mentioned that, due to the earthquake in 1976, most of the salt kitchens were destroyed due to their structural composition. Of the mentioned salt kitchens, at this moment it is hard to know their location due to the loss of the collective memory. However, the 19 found present a particular distribution and relationship with the beach space, which clarifies how they interacted with the beach to some extent. Figures 19–21 depict the last salt kitchens standing.

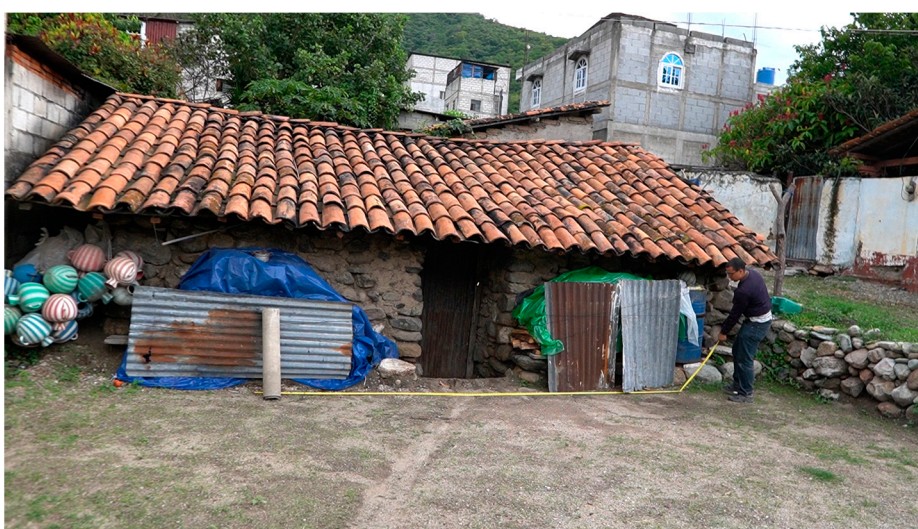

**Figure 19.** Salt kitchen No. 6. Author (2022).

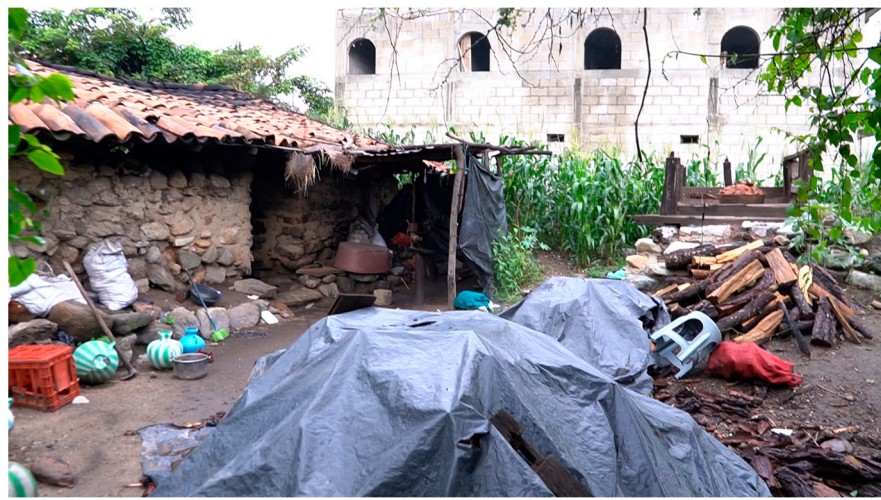

**Figure 20.** Salt kitchen No. 9. Author (2022).

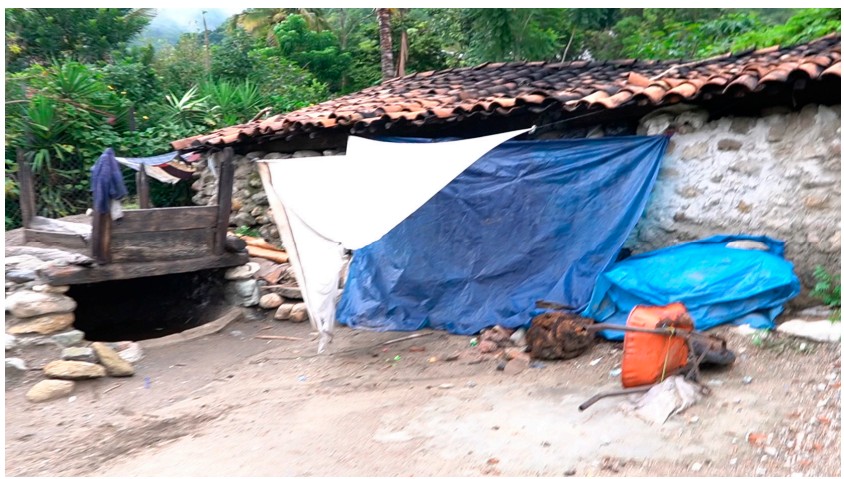

**Figure 21.** Salt kitchen No. 12. Author (2022).

## 3. Results

### 3.1. Architecture Features of the Salt Kitchen

Spanish chroniclers mentioned the architecture of the salt kitchen. The information coming from "*Titulo de los señores de Sacapulas*" (Title of the Sacapulas Lords) determined that the pre-hispanic name of Sacapulas comes from the name "**Tuja**" or "**Tujalá**." As mentioned before, Van Akkereen mentions the possibility that tuja refers to the low houses where people worked the black salt. This supposes that this architectural element has existed since before colonial times. However, the kitchen analysis presents both colonial elements and construction systems. This is defined as masonry walls: stone and mortar of semi-regular shape with a gable roof of wood structure. On the roof, a Spanish-type clay tile is used. The selection of these materials assumes the original function of the kitchen was maintained. However, its construction has been modernized at least once. This is because both the tile roof and the walls show no similarities with the pre-colonial Maya construction types; the most typical system of Mayan construction of houses was based on load-bearing walls and vegetation cover. However, this type of Mayan construction is the hardest to obtain information on, according to research conducted by Gaspar Muñoz Cosme and Cristina Vidal Lorenzo [38].

Upon analyzing the disposition of each element, it is evident that the arrangement of the materials used on each element is irregular. The walls of the kitchen, for example, are made of stone and mortar. However, the stones are selected for size rather than

shape. This results in a certain degree of irregularity, reducing the structure's sturdiness, which translates to a high probability of collapse. MSS-A mentions that several of the buildings were affected due to the 1976 earthquake, demonstrating the construction's fragility. However, there is some sort of arrangement of the stones on the front wall, specifically near the entrance door. Meanwhile, in the exterior, stones delimit the area of the outside patio. These stones do not present regularity and are only selected for their size. Within this perimeter, the space serves as a storage area for the equipment used in the salt cooking. It is also the space where the distillation basin is found. The arrangement of the basin does not follow a rule among the three kitchens, and it is assumed that the location is decided simply by the proximity to the salt kitchen. All the kitchens analyzed have in their outside patio the *"tinajas"* used to carry the thermal waters to the basin, and the *"ocote"* (pine) firewood used in the salt burning. Around the kitchen, there are also Spanish clay tiles used to exchange used tiles deteriorated from the heat inside the kitchen as well as the climate of the region. Figures 22 and 23 depicts the clay tiles location.

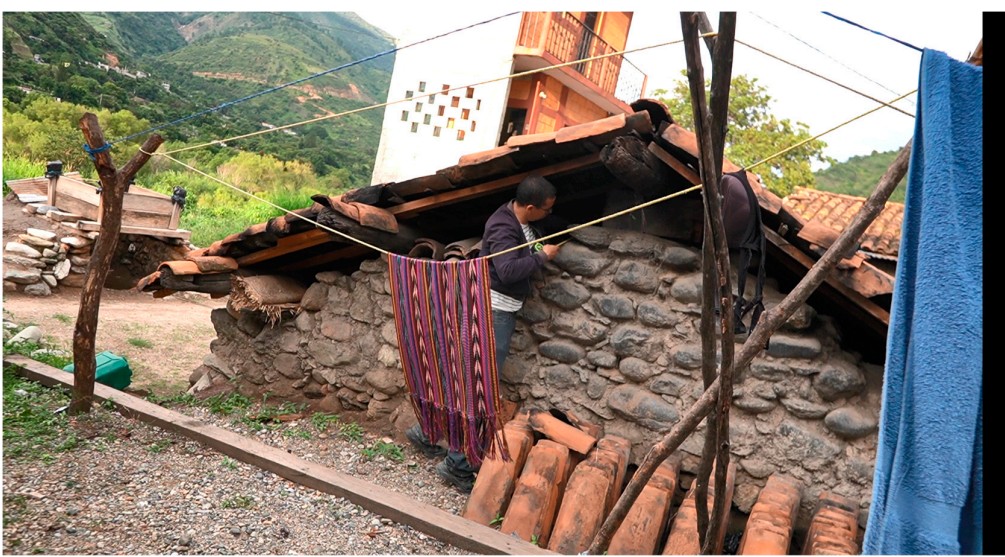

**Figure 22.** MSS-2 salt kitchen; the wall made from stucco and stones can be seen. It shows a rudimentary construction process without any internal structure. Author (2022).

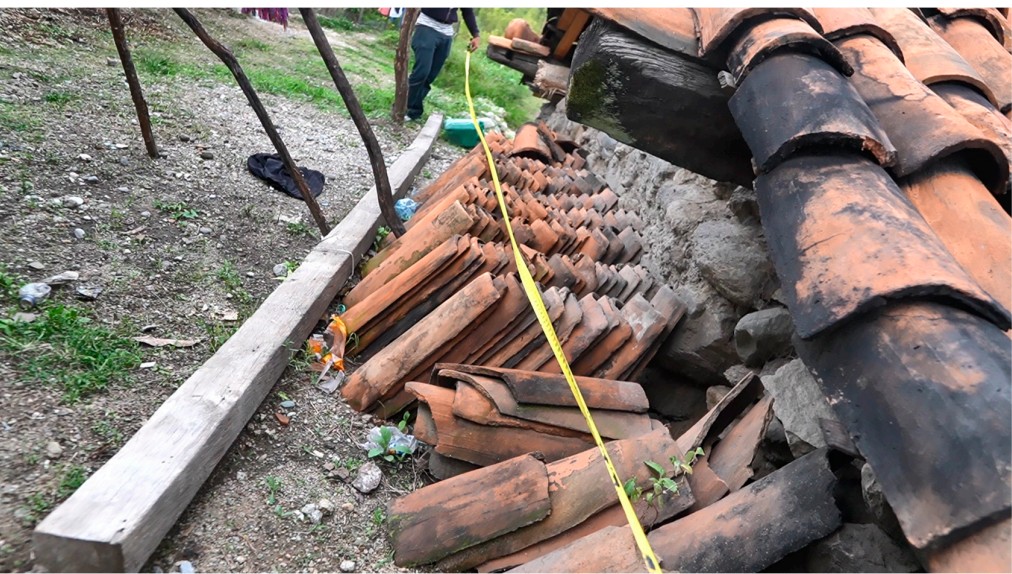

**Figure 23.** The Spanish roof tiles replacements can be seen next to the kitchen due to space constraints. Author (2022).

The salt kitchen analyzed (kitchen No. 6) has a semi-regular roof structure. This means that although it presents a typical wooden ceiling, its structure does not present regular spacing between elements. Specifically, the roof is composed of three beams and one ridge beam. Two of the three beams are circular sections, while the remaining one is a square section. The reason for this variation has yet to be discovered, but it can be assumed that it is due to the use of whatever material was available to the owner. The ridge beam rests on a large rock with a wedge in the upper area to fit the structure. All beams extend about 0.55 m outward on the edges. Because wood tends to rot due to external factors, the extended areas are covered by Spanish tiles. Following the beams are the rafters and battens. The rafters' layout is regular, with a spacing of 0.50 m between each end. The battens do not have a regular distribution, varying their spacing distance between rafters. However, it was possible to define between 5 and 6 battens arranged between the rafter spacing. The last layer of the roof is the Spanish tile. The tiles are 0.46 m by 0.28 m, and have no unique characteristics. When MSS-A was asked about the number of tiles found around the kitchen, he mentioned that due to the constant deterioration, a large sum was needed at all times. According to him, it is common to replace them every two or three years. Figure 24 depicts the wood frame, while Figure 25 depicts the front of the kitchen and location of the "*tinajas*".

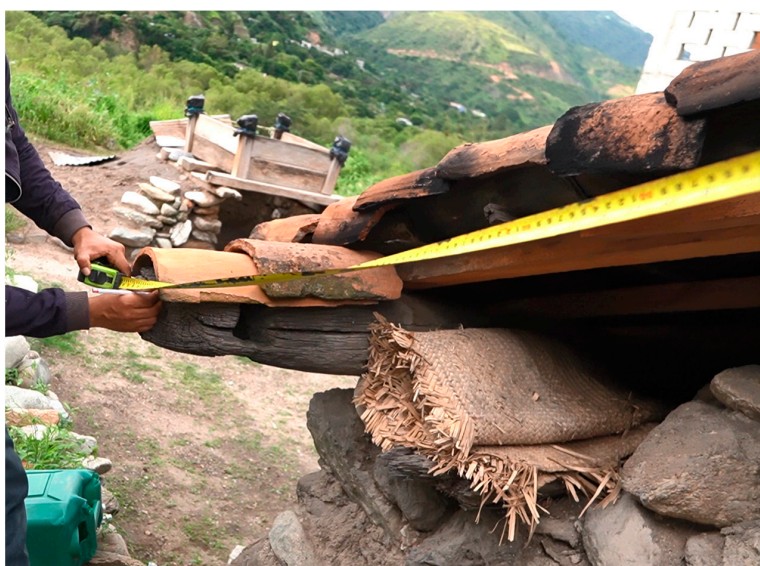

**Figure 24.** Salt kitchen extended roof structure, covered by Spanish tiles. The wear can be seen due to the black spots from both moisture and constant cooking inside the kitchen. Author (2022).

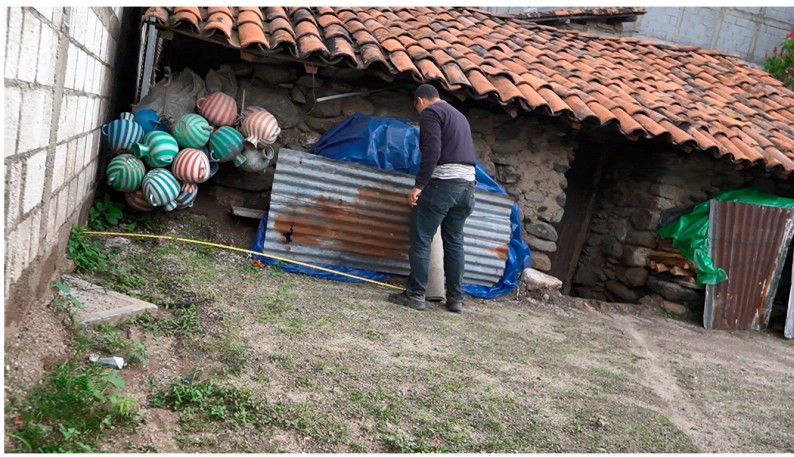

**Figure 25.** Front of the kitchen where the "tinajas" can be found, on the left side.

### 3.1.1. Interior of the Space

According to the interviews conducted and the literature research, the salt kitchen serves primarily as an encapsulation of space for cooking salt. This still holds true up to this time, as the process of salt making has not changed at all. The salt kitchen interior has no finishes on the walls or floor, which is made of earth. In fact, the interior of the space is relatively simple and has no significant qualities, being an enclosed space during the salt cooking process due to the temperature difference between inside and outside, which can "crack" the mold used to make the black salt. The result is an area that can maintain the high temperatures of the flame constantly. There are openings between the walls and the wooden beams, previously mentioned as a storage area for various materials used in the process, which can further be seen in Appendix A. This is the only area where fresh air can enter or leave the space indirectly. As mentioned in Section 1.4.7, the heart of the fire is inside the space, which must be kept in flames the entire time the black salt is made. There is no information about the temperature needed to cook the salt, which assumes this knowledge is empirical. However, the interviewees mentioned that wood from "*ocote*" (pine) is needed because it "burns better," strengthening the hypothesis that this knowledge is not scientific.

Inside, the "heart of the fire" is created: a rudimentary oven of about 2.5 m in diameter, according to the research conducted. It comprises oval stones called "tenamastes" that raise the salt molds. Previously, the use of 40 of them was mentioned. However, the result of the research was that nowadays, 15–18 tenamastes are used due to the reduction in salt production. This results in a much smaller "heart of fire" than reported by Reyna and Monaghan during their research. The size of the heart seems to be governed by its functionality and convenience to the salt maker. During the observed process, it was identified that eight salt molds are made per burn. The function of the salt maker within the space is to maintain this flame at a specific temperature, introducing "*cargas*" (loads) of firewood into the fire during a process that lasts about 6 to 8 h.

There are some discrepancies between what was mentioned in the interview and the overall function of the salt kitchen. Specifically, it was determined by the interviews that the space is used mostly a warehouse saving soil impregnated with salt. According to MSS-A, this soil will be used throughout the whole year, even the wet season when it is impossible to impregnate the soil with salt. This in turn determines how economically flexible one salt maker would be during the whole year, as more soil equals more potential economic benefit. In fact, the salt cooking process presents a rudimentary way of cooking, effectively transforming the space into a stove. This process is something that both salt makers and ex-salt makers mentioned as a cause of people's disinterest in continuing with this tradition. As it turns out, the subject cannot open the space until the end of the cooking, a process in which several noxious gases are generated.

### 3.2. Relationship between Architectural Elements and Space

Beyond the number of kitchens, essential characteristics of how the architecture interacts with the space were found. During the field visit, it was possible to define three significant areas represented in Figure 19: A (current salt beach), B (intermediate plain), and C (land near the "pit"). General pictures of the space are found in Figures 15–18. Area A is currently used, Area B is a plain used for cattle, and Area C is abandoned. Two areas (A and C) are connected by main roads (pink). The three areas are also connected by a makeshift road created by the people of Sacapulas (yellow). This road also connects all the spaces of the salt kitchens. The salt beach is connected by two roads, which are unsafe due to neglect. However, they are the only means of access. Finally, areas A and C have roads that do not connect to any place (green color).

At first glance, it could be hypothesized that the distribution of the salt kitchens is a function of the proximity to the salt beach, as it was impossible to determine specific factors of their construction other than the proximity to the area. Two interviewees mentioned that several kitchens existed close to each other because they belonged to the same family. In this case, kitchens 2–4 and 6 belonged to the same lineage of salt makers. It is unknown if this also happened in other salt kitchens or if there is some relationship resulting in the construction of the architecture. Figure 26 shows the current location of the salt kitchens and roads that connect to the different spaces. Currently, only groups 1–12 are close to the beach. Groups 13–15 are closer to Area B, and groups 16–19 are closer to Area C. Specifically, the roads that connect to Area A (blue color) are relatively close to the existing salt kitchens. Although several salt kitchens were destroyed due to external factors, others were destroyed due to a lack of people interested in salt. While several of the spaces have been reused, many of them are abandoned. Figures 27–29 show the actual situation of the spaces.

As resumed in Table 5 of all the salt kitchen areas, only No. 3, 5, and 15 were reused for planting (16%). No. 7 and 8 were reused for new construction (10.5%). No. 7 was used for constructing a private house, while No. 8 was reused for constructing a public, synthetic grass soccer field. This new use is probably due to its location in front of a main street compared to the other kitchen spaces.

If it is assumed that the salt beach has always been Area A and its surroundings, questions arise about its accessibility. Considering kitchens 13–19 had access to the beach, it would be challenging for a salt maker to travel from A to the salt kitchens as it demands too much energy. Therefore, the location of these kitchens is not logical if it relates to Area A. In addition, Area B is a plain, and has a direct connection to the Chixoy River; this is important because it does not present fences like other terrains next to the area: this space must not have a current owner. Its topology supposes it was an access road to the area in front of the Chixoy River. This is supported by the informal roads found in both A and C. Area C especially presents a connection to the town of Sacapulas, utilizing a main road and two informal roads to the north and the west. The northern road connects to the outskirts of the town of Sacapulas. Meanwhile, the road to the west only connects to somewhere currently buried by the river. A previous connection to some existing area in front of B must have existed. However, this area is currently inaccessible due to the Chixoy River.

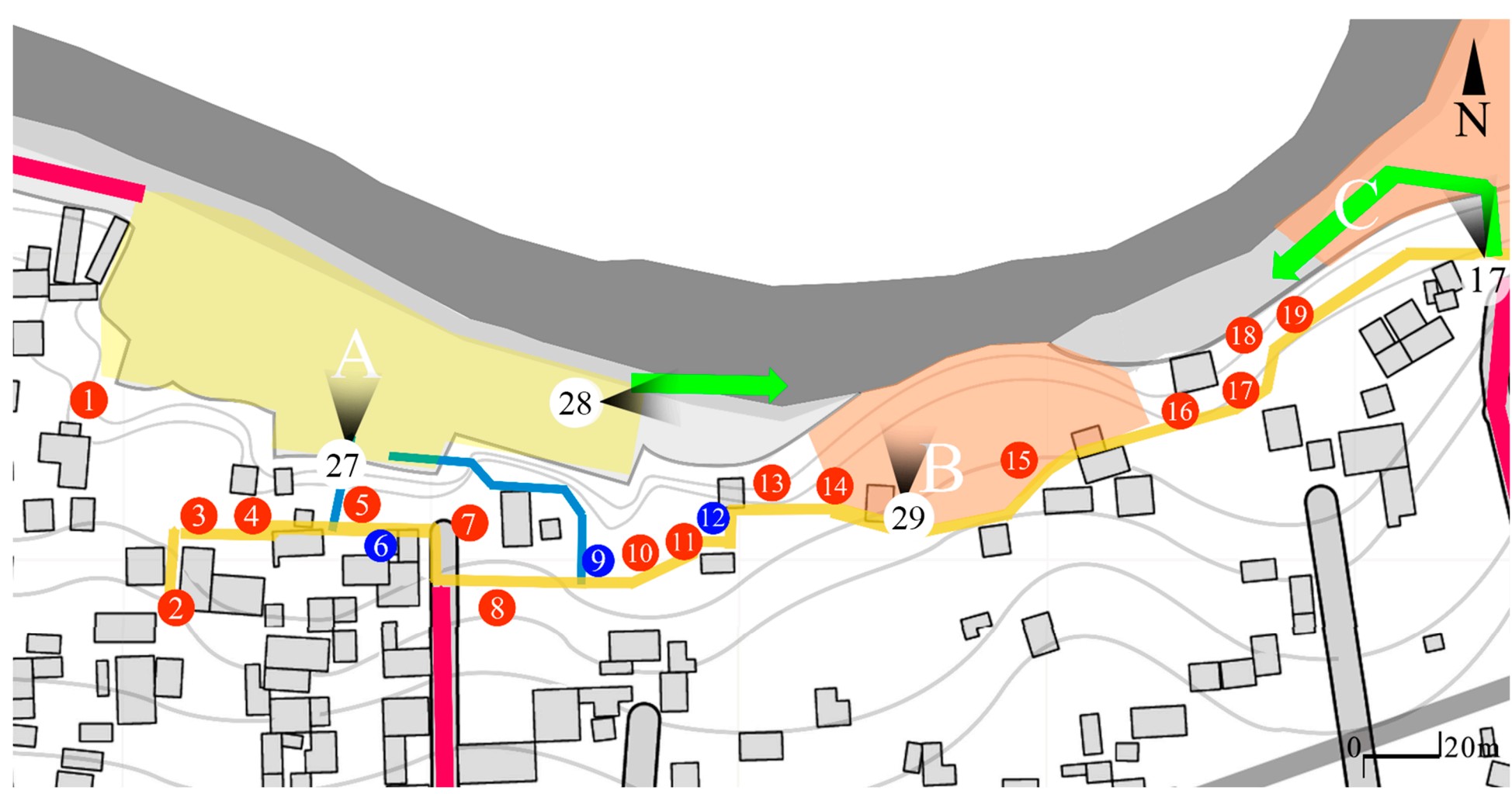

**Figure 26.** Area analysis. In red, destroyed kitchens. In blue, existing kitchens. In white, location of the pictures taken (Figures 27 and 28).

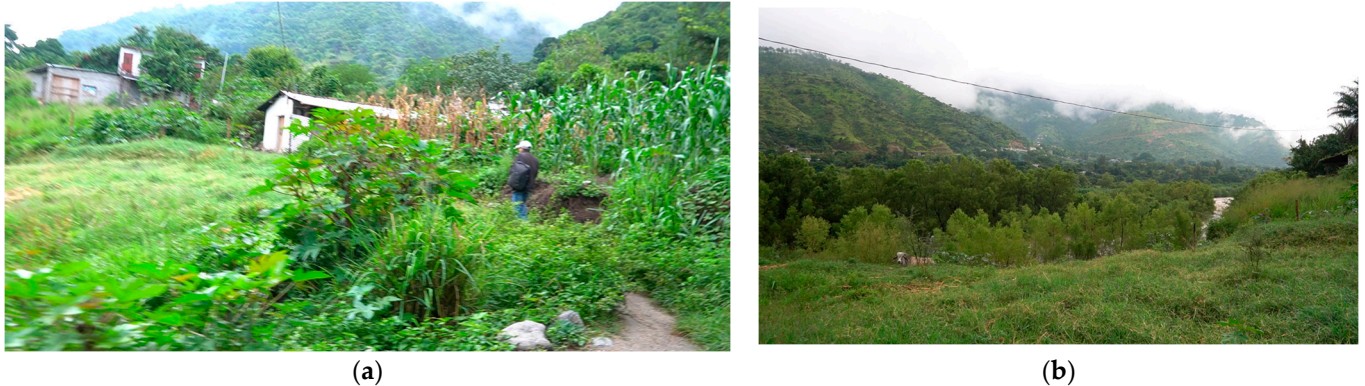

**Figure 27.** Pictures from Area A and its current state. This is the actual "black salt" beach, as it is known. Author (2022).

(**a**)     (**b**)

**Figure 28.** Picture (**a**) represents the road going across Area B. Picture (**b**) presents the current state of Area B. Author (2022).

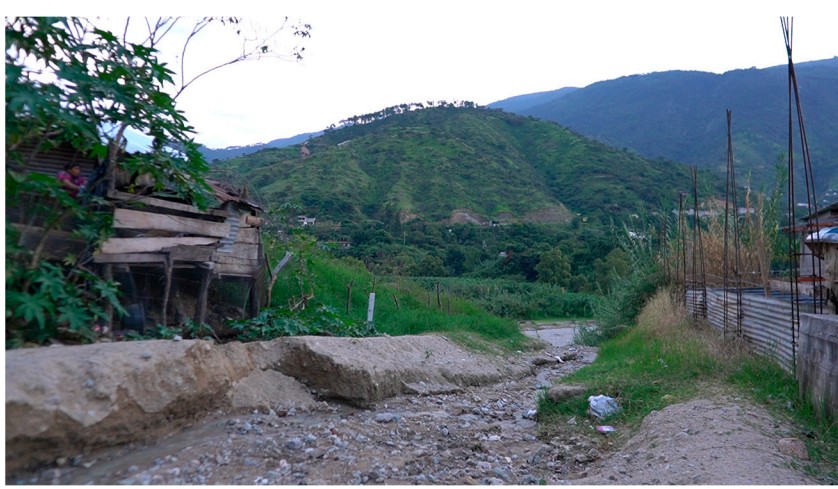

**Figure 29.** Area C. Author (2022).

## 4. Discussion and Conclusions

### *4.1. Salt Kitchens, Their Function, and Current Issue*

Continuous natural disasters and lack of interest have impacted the traditional Mayan knowledge of black salt. This research successfully created the first planimetry of the traditional salt kitchen, safekeeping its knowledge for further research and understanding. It also analyzed its relationship towards the space and its current usage. Salt kitchens are a rudimentary way of creating salt. This was confirmed by two crucial factors: the research conducted by Andrews [26], and how the salt kitchen functions. In general terms, the kitchen shows evidence of being fit for local people rather than outsiders. The salt kitchen presents measurements that only seem logical if we consider the height of a person is around 1.60 m. Considering that the height of people in Guatemala is around 1.60 m [39], it does not follow an international standard, rather a local one. This suggests a much greater importance beyond the black salt industry, and being connected to the people and culture of the town.

During the analysis, it was found that the function of the kitchen is mostly as a space for storing earth impregnated with salt used during the off season, representing an annual economic income. Without presenting a specific aesthetic, the space is a highly functional area, as the bigger the internal space, the more money salt makers could get during the year. Considering the price of maintaining the structure and salt cooking materials, salt farmers are encouraged to resort to better incomes. Evidence of this is that many ex-salt makers preferred to demolish the kitchen and turn it into a crop-growing space. By destroying the salt kitchen, the possibility of creating salt is also destroyed. It also breaks the generational tradition that this specific family has been performing for years. Although the loss of interest in black salt has been a progressive event due to both human and natural factors, it is the 1976 earthquake that was the determining factor in the loss of the industry. According to MSS-A, most of the salt kitchens were destroyed in this event due to the construction system used in them. Due to the lack of internal structure, any natural catastrophe could destroy the kitchen, earthquakes being cited as a common occurrence. Considering the regularity of these natural events and their magnitude, it is self-evident that the kitchen slowly deteriorates to a point where structural fractures develop.

The reason why the structural system was not modernized up to this point was unknown to the interviewees. However, it can be hypothesized in terms of two reasons:

- Age of the constructions: When interviewing the salt makers, it was evident that the exact date of construction of the salt kitchens is unknown, as they are inherited among family members. Taking into account this, together with the construction system used (adobe masonry construction), and the lack of mention by Spaniards in their chronicles, it is hypothesized that the date of the construction was at least during the colonial era (1524 onwards).
- No new kitchens built: Due to the loss of interest in the industry and personal economy of the salt makers, there are no data on new salt kitchens built. The industry depends on the three existing kitchens to subsist, resulting in an extremely delicate sustainability and resilience towards any natural or man-made catastrophe.

Creating black salt also harms the salt maker, and is a process yet to be modernized. Instead, the same tradition has been maintained since before the conquest. The salt makers understand that the process is highly body-consuming, but there has not been any action or interest in improving it. When asked for their opinion, all of the salt makers mentioned that it is the "*costumbre*" (social custom) that dictates how salt is made, regardless of its negative impact. It can be hypothesized that the lack of development resulted in an industry that is not very competitive in the market and a cultural aspect at risk of complete loss.

### 4.2. Location and Relationship with the Black Salt Beach

Considering that there are no records of the salt kitchens and planimetry of the space, it can be hypothesized that the original shape of the salt beach was different from the current situation [30], to such an extent that the beach area was much more prominent to the north than it is today. The location of kitchens 13–19 is more logical if one assumes an area in front of B, connecting Area A to C. The chronicler Martin Alfonso Tovilla makes a mention regarding the size of the black salt beach in his recount [29]. Hypothesizing that he meant that the total area was around 0.687 kilom, by measuring the land from Area A to the east, this results in an extension slightly in front of Area C, towards the mouth of the Chixoy River. If we take a hypothesis of how the original river was, an affluence on the river flow would mean a progressive crooking of the river towards the black salt beach. This is due to the area being at an extremely low height and being primed for modifications by the river. Due to the shape of the current meander located west of Area A, the increase in the river current could result in slowly burying Area B. The hypothetical original shape of the beach is presented by Figure 18. There are still new questions regarding this hypothesis:

- What was the size of the area in 1979, when Ruben E. Reina and John Monaghan conducted their research in the area? Although they took photos, only one presents the black salt beach, and the shape of it is barely visible. In their research, there is mention that the floods in the 1940s and early 1950s buried most of the beaches, while only one part was unburied. It also mentions that the salt makers informed them that only one-fifth of the original space was being used. However, they do not mention where the other four parts of the space were located.
- What was the size of the area in 2002, when Ordoñez Chicano conducted his research in the area? Although he took a photo, his research never delved into the exact measurements of the salt beach. There is no mention of the size during that year or before the floods. Regarding floods, he mentions one in 1949 that buried half of the present salt flats. Like Reina and Monaghan, there are no documents that can determine the total size of the area.

In order to obtain information regarding the buried beach, it is necessary to analyze the composition of the currently buried soil. However, this is beyond the scope of the research carried out.

The results show a close relationship between the architecture distribution and the beach space. If we take the original area of the hypothesis, the relationship between the urban fabric (which does not present urban planning), the shape of the beach, and the black salt industry is evident. Proof of this is how the salt kitchens were distributed in space, with the urban fabric arranged with the need for access to the kitchens. However, all of the roads connecting the salt kitchens are informal, both in shape and materials used (earthen, defined specifically by function rather than aesthetic or safeness). This implies lesser interest from the Sacapulas government, resulting in little investment towards the access roads. The only part of the road that uses concrete is the steps that connect the southern area where the salt kitchens are located, with the existing space of the black salt beach [24]. However, evidence shows that the measurements of these elements are slightly irregular and are not in an optimal state. Considering that the people dedicated to black salt are at an advanced age (between 50 and older), this poses an extremely high risk to their safety. Moreover, transporting the black salt to the kitchen proves a difficult endeavor for someone of this age.

The location of the kitchens has helped them be sheltered from any flood or natural disaster occurred during the years, as they are located in elevated terrain. However, the loss of salt harvesting areas is the main reason the salt industry shrank considerably, affecting their usage. In addition, the difficulties of preparing black salt have alienated new generations. The interviews showed that many people prefer to repurpose the kitchen spaces into crop-growing areas, as it is much more profitable and demonstrates an alienation from the Mayan customs rooted in the black salt. Today, many kitchen spaces are abandoned or within large plots of land belonging to the children of the original salt maker families. However, the urban thread and salt kitchen locations present a history where this was different which is depicted in Figure 30. An era where the people, the salt workers, and the architecture were important aspects of their identity.

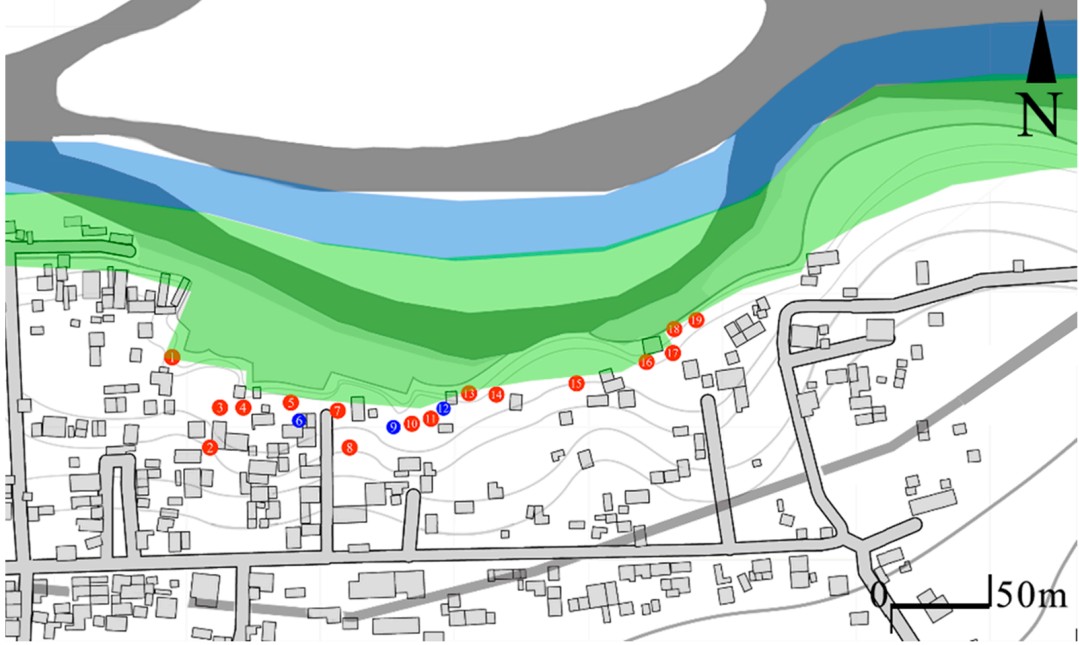

**Figure 30.** The hypothesis of the original black salt beach. Blue: Chixoy River, Green: salt beaches hypothesis. Red defines the destroyed salt kitchens, in blue, the existing ones.

### 5. Conclusions

By taking into account the research conducted since Andrews and the latest researchers, it is understood that the local salt industry is effectively dying. As discussed in Section 1, Andrews mentioned certain causes that will eventually lead to the loss of the black salt. This and the previous research conducted on the subject have confirmed it to be true, and also identified new reasons that have accelerated this loss. As a consequence of the industry dying, cultural expressions such as the architecture have also been affected. Black salt is a product with close ties to the people of Sacapulas. This has been evidenced by how the black salt beach has influenced the architecture, town, and urban fabric. In addition, Maya texts and chronicles present an era when the native people of Sacapulas celebrated their roots and black salt work in religious demonstrations [31], evidence of the importance of the space. Today, this is different; the celebrations are no longer held and a part of the culture has effectively died. However, a specific group of people still practice this profession. By developing this research, a documented account of the architecture has been successfully completed. Now, several new steps must be taken to properly understand to what extent the salt beach space and industry can be recovered. How we can adapt such traditional methods to make them economically feasible, or at least interesting enough for the people, is an important point to discuss. Plenty of research has yet to be conducted in the area, which could yield a better understanding of the current intangible value it presents to the people of Sacapulas. By developing the current research, it was found that people outside of the salt industry still use the beach, even knowing that the space presents an actual danger. A first step should be to understand what the usage and frequency of the salt beach are, and what activities are currently conducted in the space. This research will yield the necessary information to decide what the best direction to recover the space is, and the feasibility of such action. Information about the usage outside of the salt industry does not exist yet, and no research conducted up to this point has ever been interested in activities other than the salt creation. There is an important need due to such historical lack of information, and prompt action is required.

After understanding its current usage, future research may focus on two aspects: the black salt product, and the recovering of the salt beach. As discussed previously, one of the most attractive reasons for the black salt is the medicinal properties attributed to it, which have yet to be proved by research. This goes beyond the expertise of this researcher, yet the subject might prove attractive to researchers interested in Mayan culture who desire its safeguarding. Meanwhile, this research has hypothesized the original shape of the black salt beach due to the relation between known architecture elements, urban thread, and the current black salt beach. This should be further researched as research and locals suggest that more salt beaches existed beyond the ones found. It is also noted that soil analysis should be conducted around the hypothesized area to understand if, in effect, the flooded area presents similar properties as the current salt beach. Beyond this, the most important decision is the call for action. The benefits of such work go beyond saving a culturally important product; it impacts the livelihood of people living in constant danger. All the research conducted in the area has mentioned this, and yet there has been a holistic approach towards danger reduction. By managing, planning, and executing, the region of Sacapulas can capitalize on the natural resources that the salt beach has given to them for a long time. And by saving part of the black salt industry, an opportunity for cultural celebration will be possible for the Guatemalan people. A proper plan to safeguard in any way the culture of the area while also helping the community can be conducted in many ways, yet the economic feasibility of said action will shape decision making. Implementing KIPS for an auto-sustainable approach, recovering the "costumbre", and cataloguing and creating urban and architectural planimetry all help raise awareness and will eventually revitalize the black salt beach.

Regardless of the final outcome, decisions are being made right now. And hopefully in the future, both the town of Sacapulas and the black salt can become an attractive and important part of Guatemala, as they were long ago.

**Author Contributions:** Conceptualization, S.M. and L.P.Y.S.; methodology, L.P.Y.S.; software, L.P.Y.S.; validation, S.M. and R.N.; formal analysis, L.P.Y.S.; investigation, L.P.Y.S.; resources, S.M., R.N. and L.P.Y.S.; data curation, L.P.Y.S.; writing—original draft preparation, L.P.Y.S.; writing—review and editing, S.M. and R.N.; visualization, L.P.Y.S.; supervision, S.M. and R.N.; project administration, S.M. and R.N.; funding acquisition, S.M. All authors have read and agreed to the published version of the manuscript.

**Funding:** This work was supported by the Japan Society for the Promotion of Science Grant-in-Aid for Scientific Research (A) Grant Number 24H00341.

**Institutional Review Board Statement:** Ethical review and approval were waived for this study since no data and information related to the ethical guidelines were at the discretion of the committee at Hokkaido University.

**Informed Consent Statement:** Informed consent was obtained from all subjects involved in this study.

**Data Availability Statement:** Data is contained within the article.

**Acknowledgments:** A big acknowledgement to the non-profit organization RAICES for their help during the development of this research. A big thanks to the people of Sacapulas who helped to conduct this research.

**Conflicts of Interest:** The authors declare no conflicts of interest.

## Appendix A. Planimetry from the Salt Kitchens

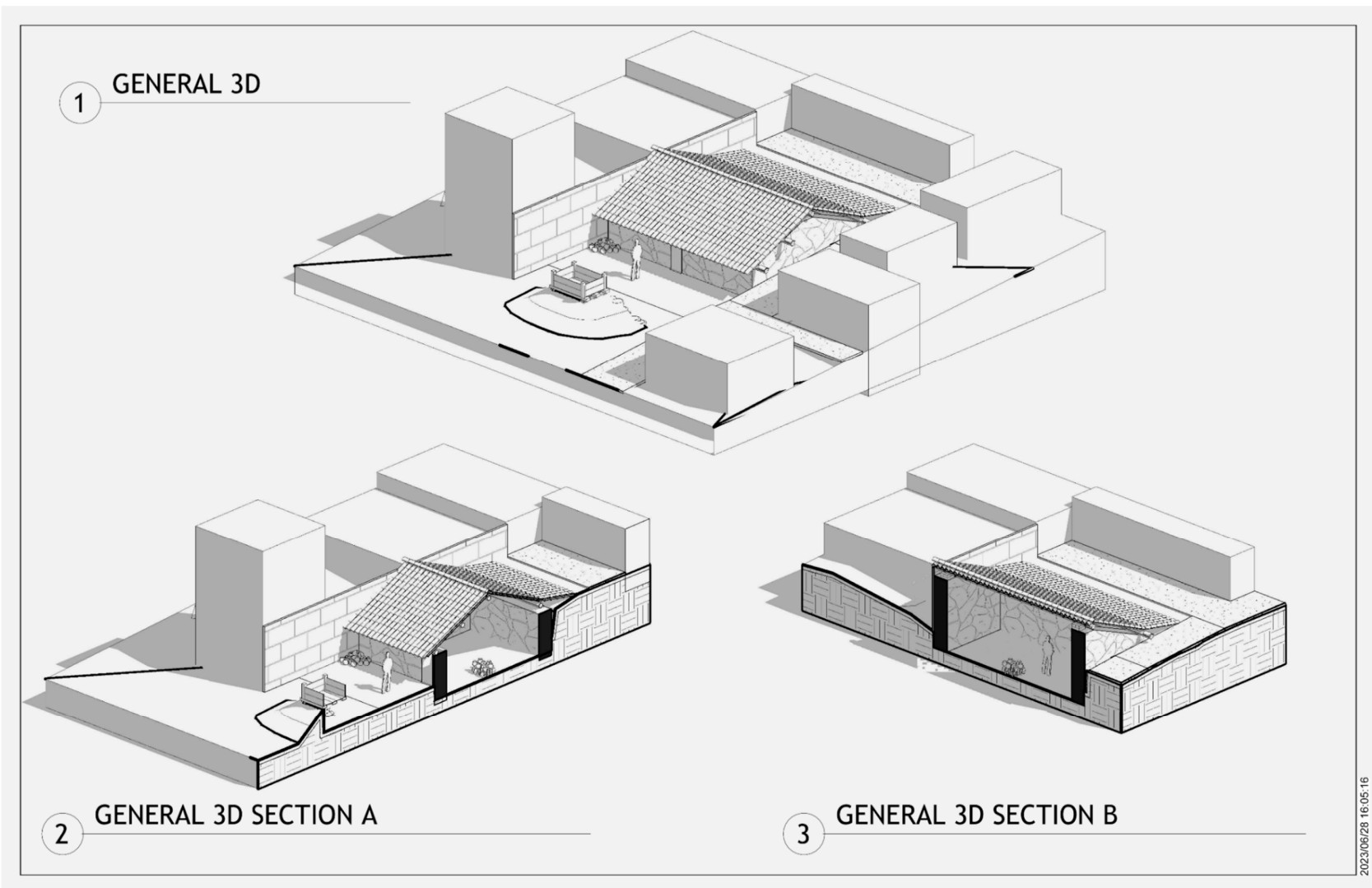

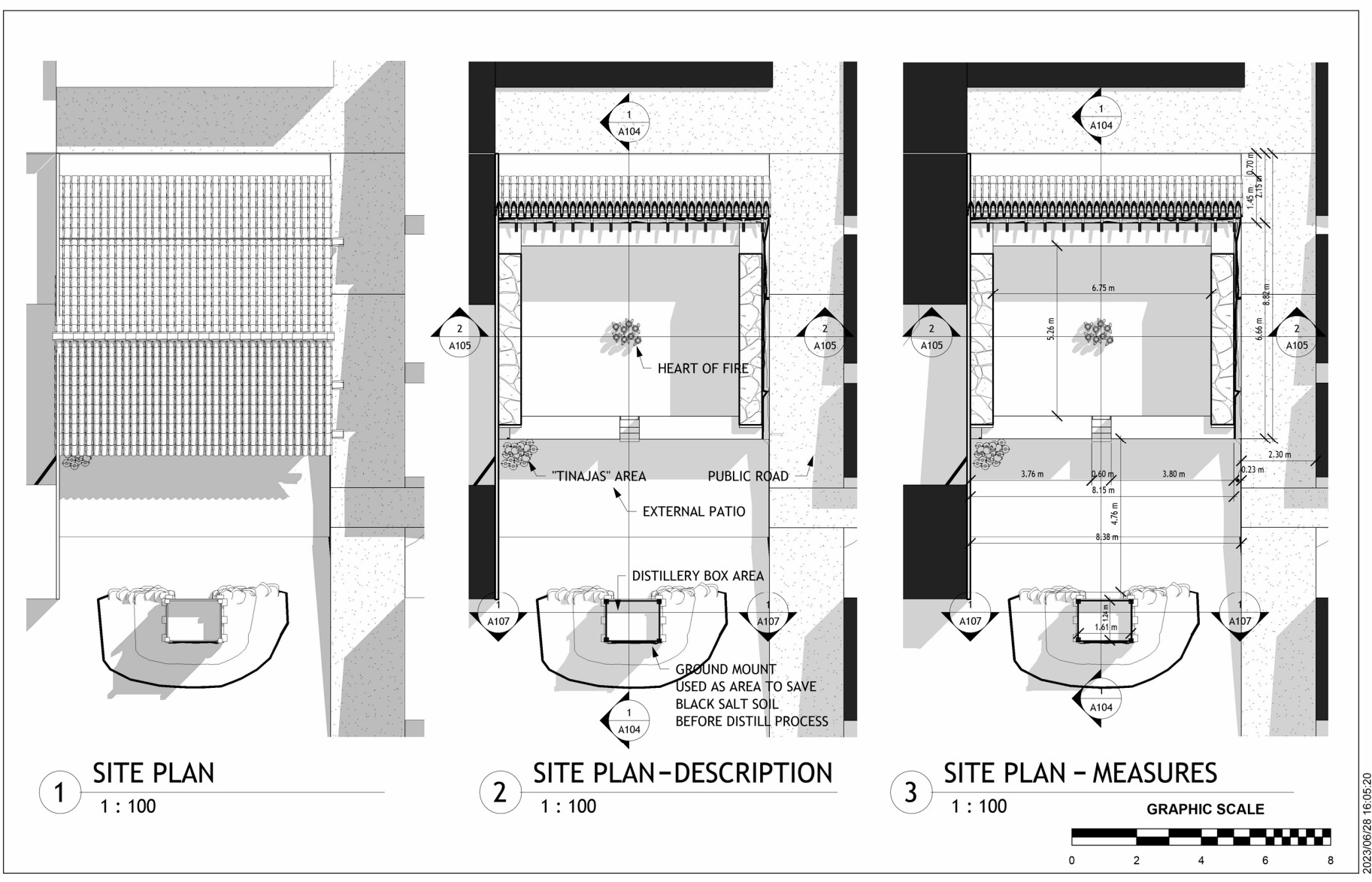

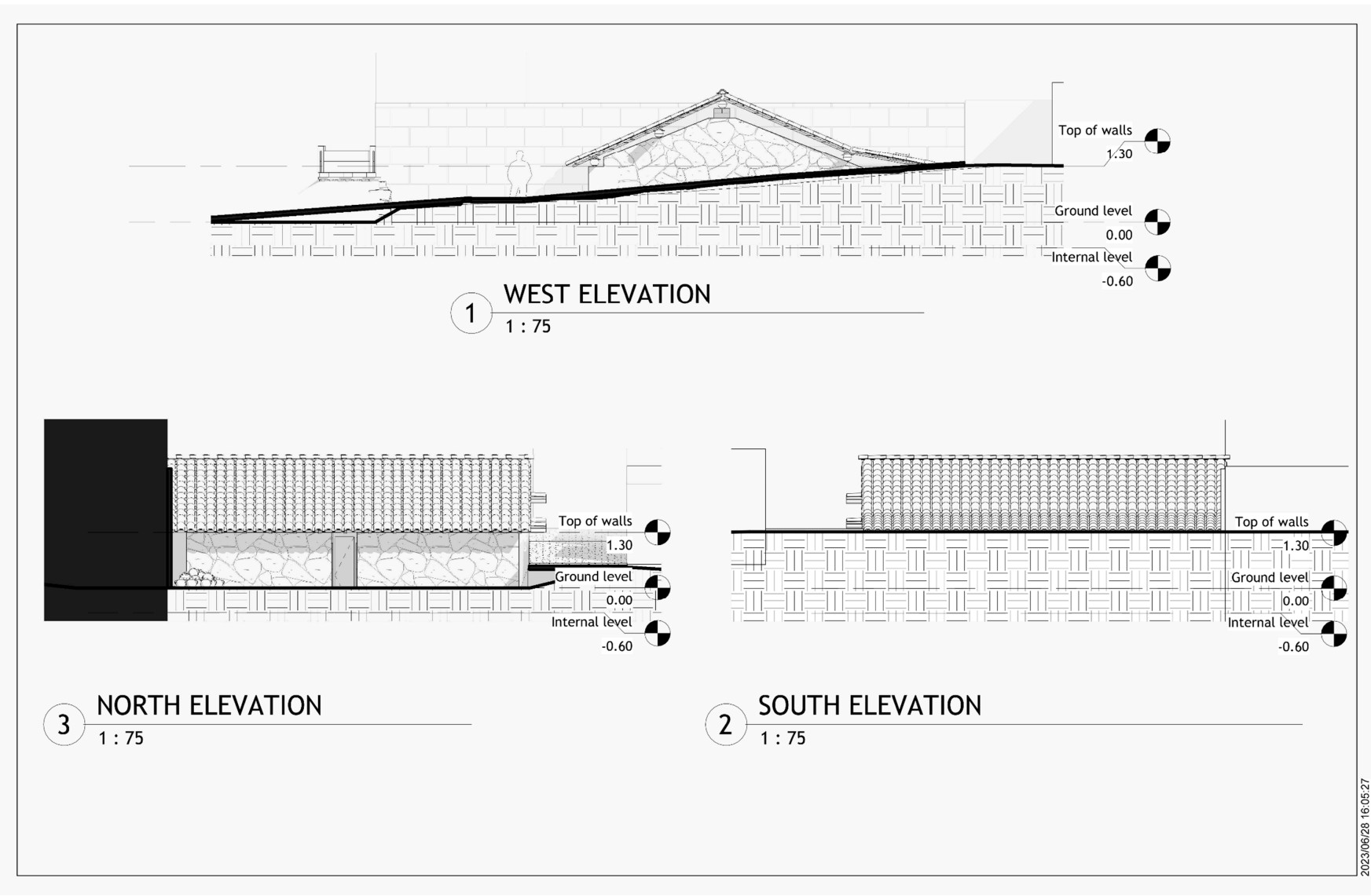

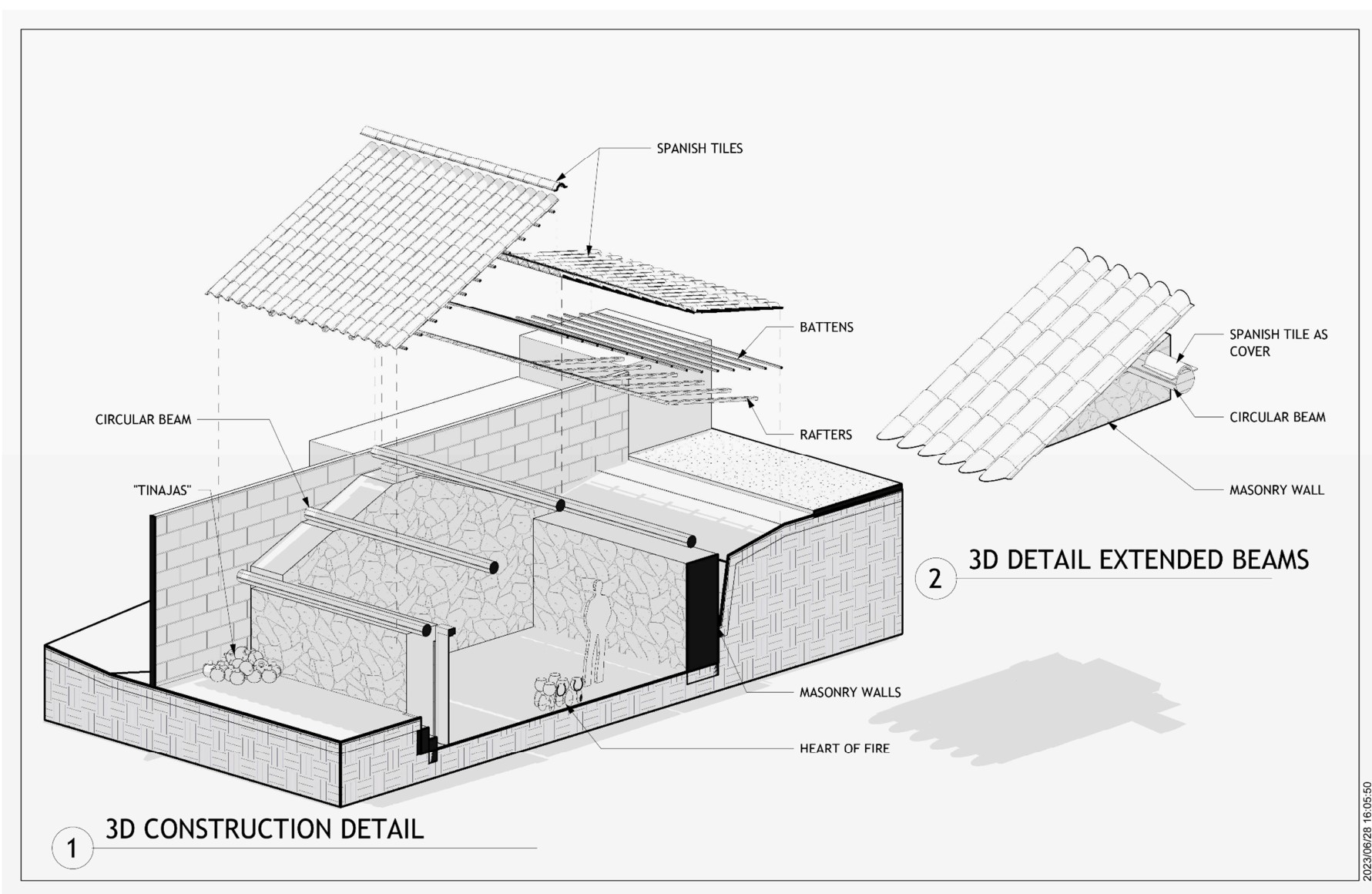

SPANISH TILES

BATTENS

RAFTERS

CIRCULAR BEAM

"TINAJAS"

MASONRY WALLS

HEART OF FIRE

SPANISH TILE AS COVER

CIRCULAR BEAM

MASONRY WALL

2 — 3D DETAIL EXTENDED BEAMS

1 — 3D CONSTRUCTION DETAIL

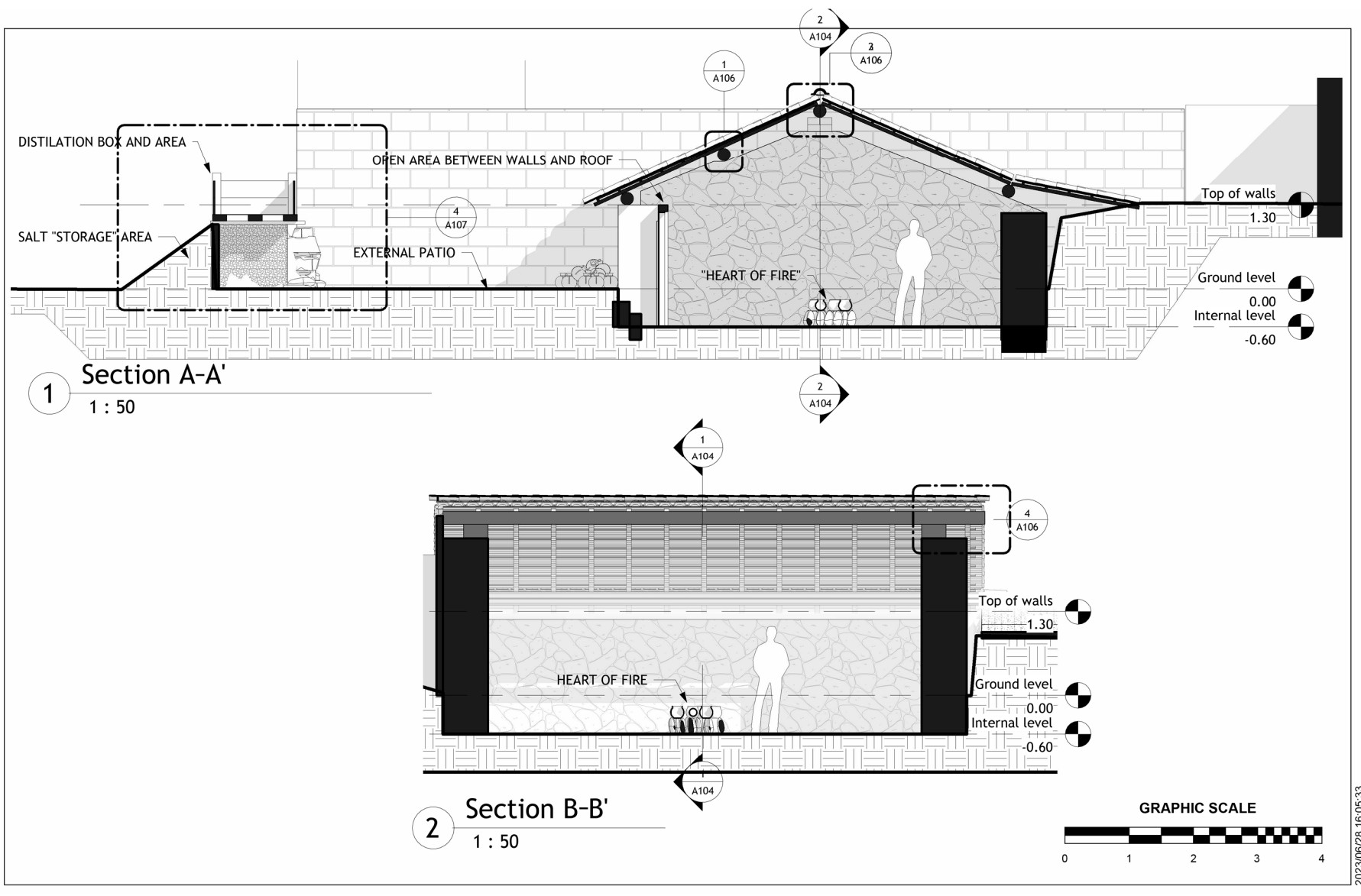

Section A-A'
1 : 50

Section B-B'
2 : 50

GRAPHIC SCALE

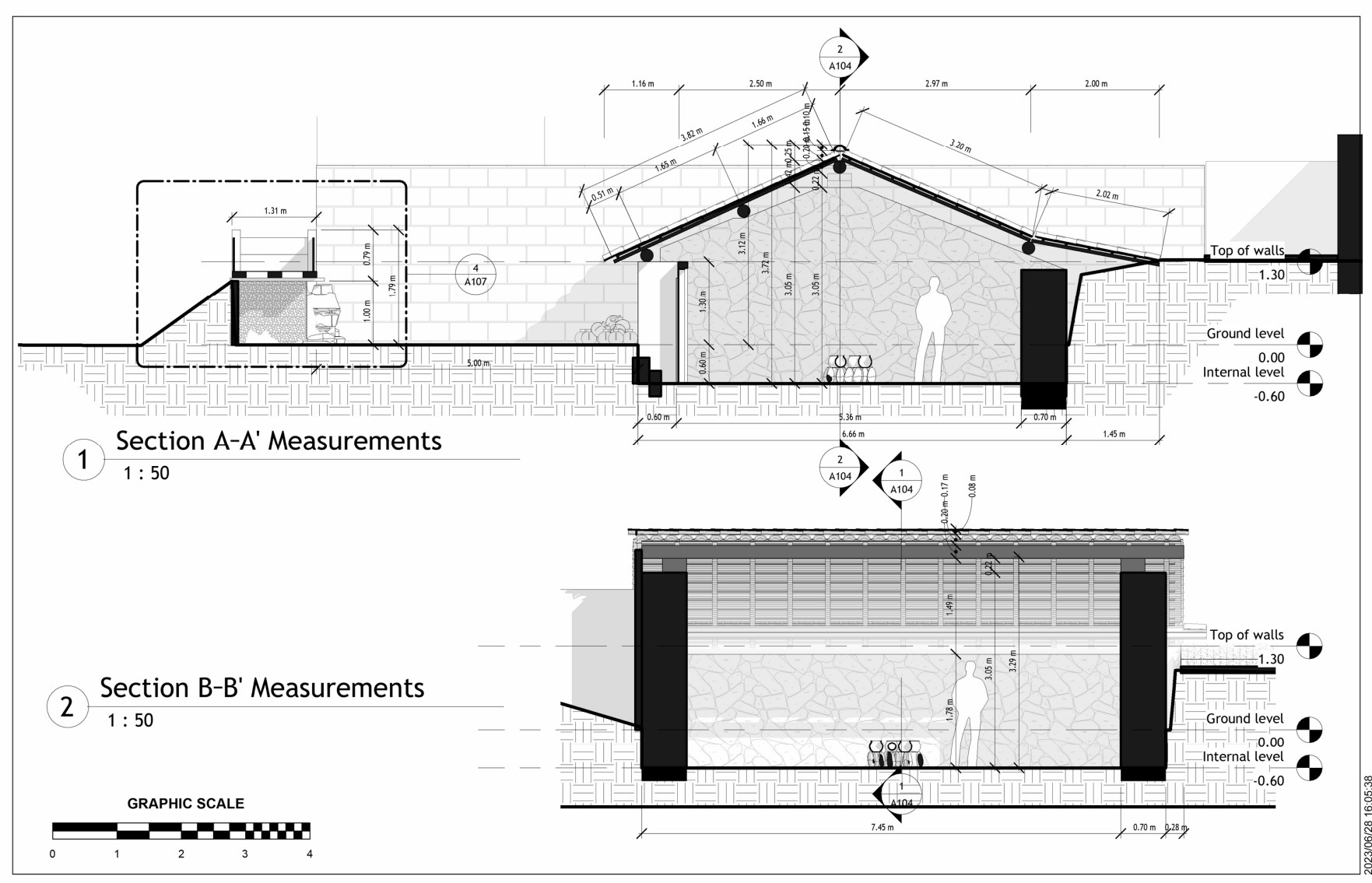

**Section A-A' Measurements**

① 1 : 50

**Section B-B' Measurements**

② 1 : 50

**GRAPHIC SCALE**

0    1    2    3    4

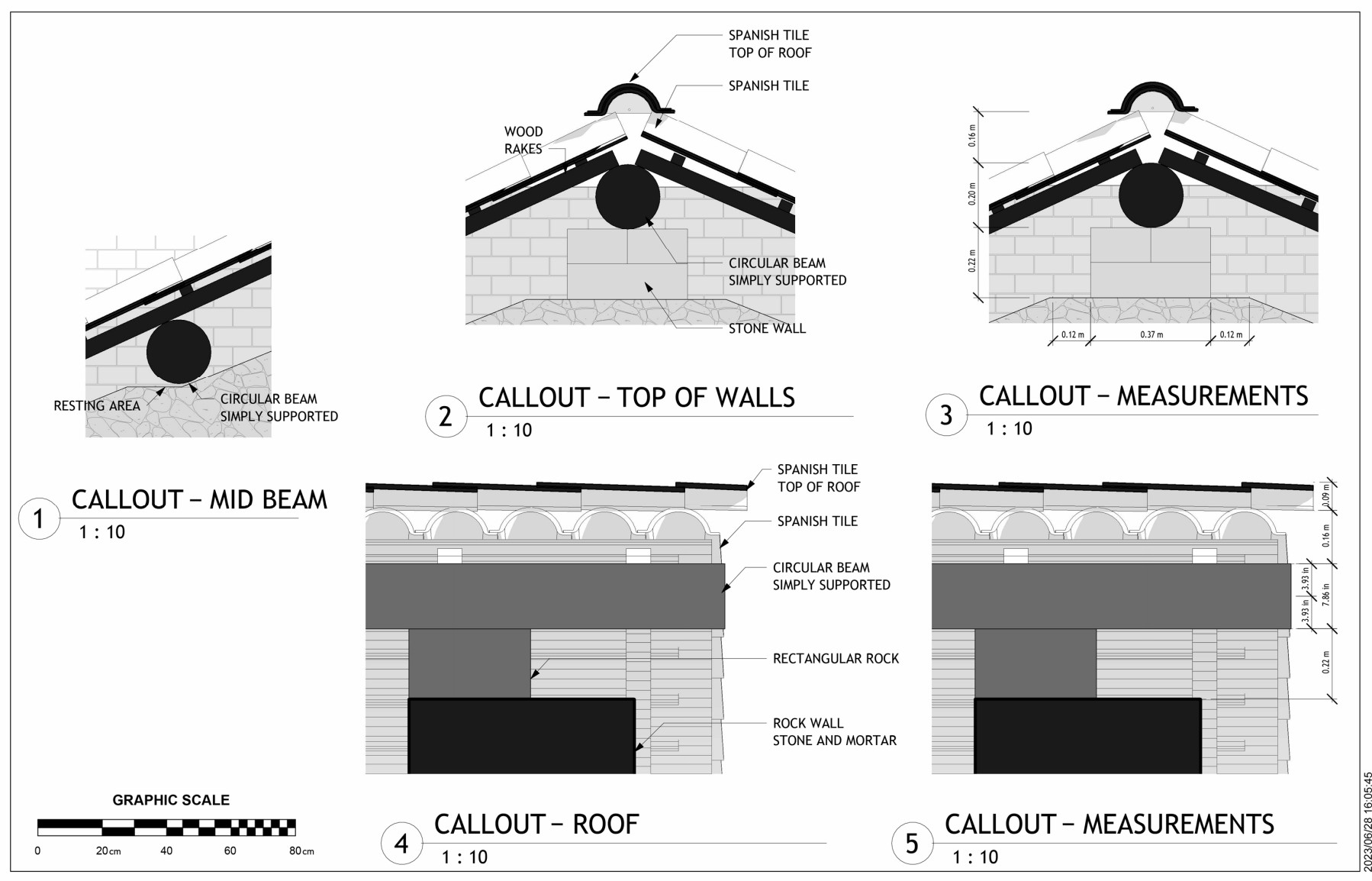

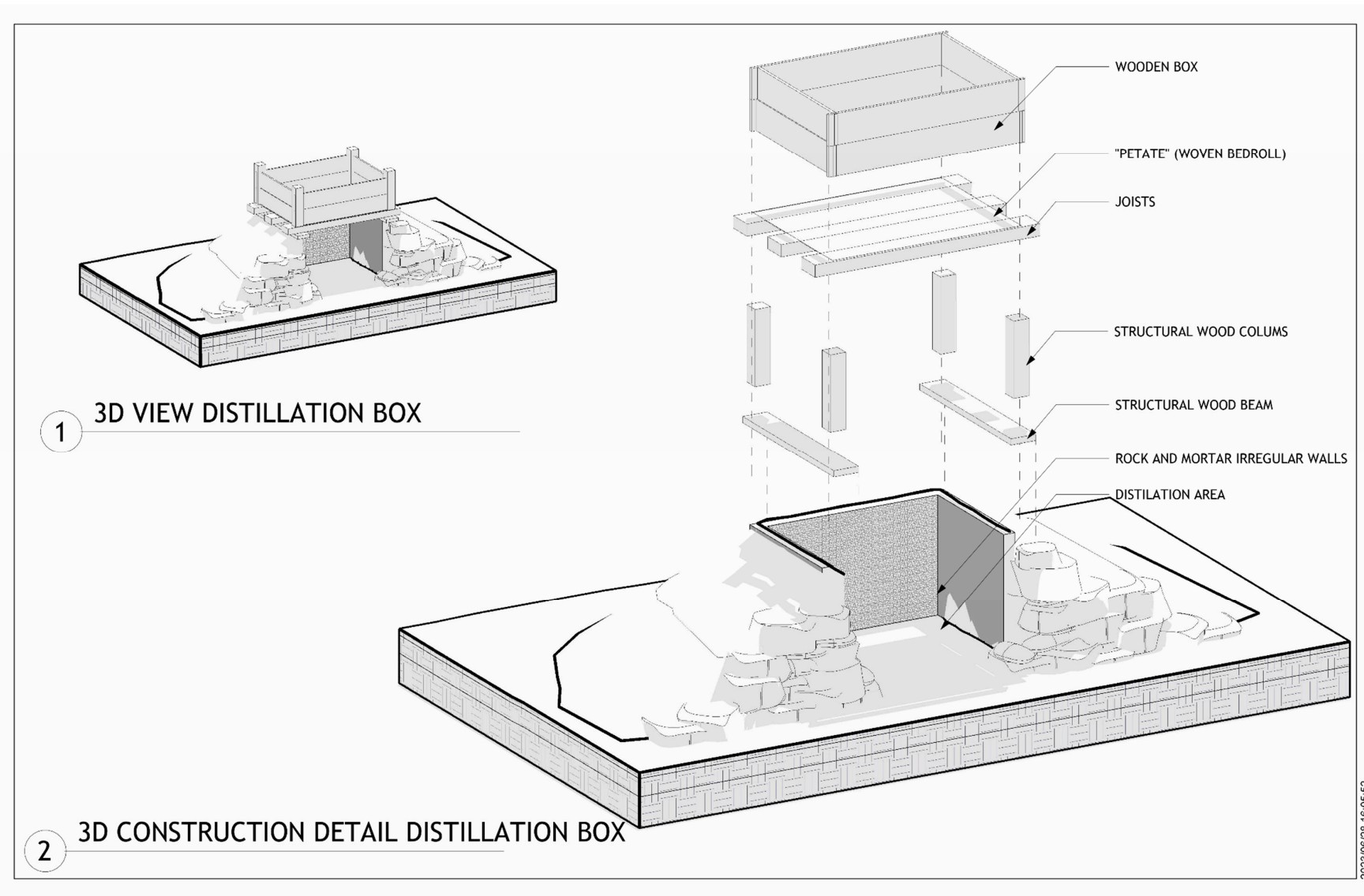

WOODEN BOX

"PETATE" (WOVEN BEDROLL)

JOISTS

STRUCTURAL WOOD COLUMS

STRUCTURAL WOOD BEAM

ROCK AND MORTAR IRREGULAR WALLS

DISTILATION AREA

1 3D VIEW DISTILLATION BOX

2 3D CONSTRUCTION DETAIL DISTILLATION BOX

2023/06/28 16:05:52

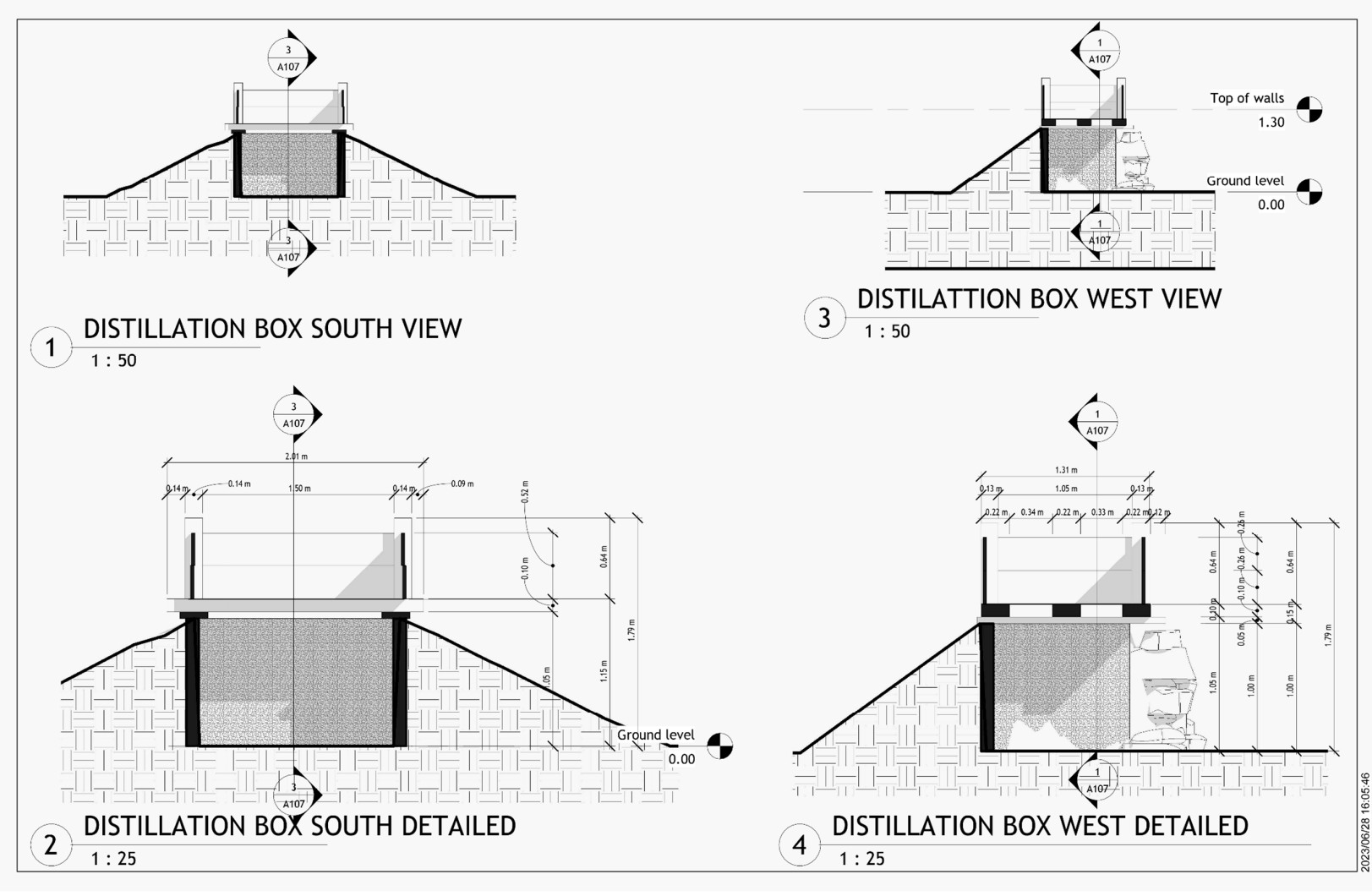

DISTILLATION BOX SOUTH VIEW
1    1 : 50

DISTILATTION BOX WEST VIEW
3    1 : 50

DISTILLATION BOX SOUTH DETAILED
2    1 : 25

DISTILLATION BOX WEST DETAILED
4    1 : 25

# Appendix B. Questionnaire Employed in the On-Site Survey

**Black Salt in Guatemala**

**The objective of the interview: to understand the current situation of black salt, the use of the space around the salt beaches, people's perception of this space, and the future of this industry.**

Therefore, the following questions are given, which are necessary to obtain the necessary data from the pre-survey. They don't necessarily have to be the specific question, but related enough to get the desired answer.

---

*For the interviewer, read the following statement to the interviewee:*

Dear Mr./Mrs. (enter name of interviewee) Nice to greet you, my name is *(interviewer name, present identification document )* , and I come on behalf of Mr. Luis Pablo Yo Secaida. I understand that according to what has been discussed with Mr. Yo Secaida, I would be carrying out the agreed interview with him. **I will proceed to read you the following clarification:**The following interview is conducted in order to obtain information for strictly academic purposes. This information will be processed in order to obtain quantitative and qualitative statistical data, and will serve as the basis for other interviews and surveys that will delve into the influence and impact of the black salt industry in the Department of Quiché. In addition, you would be requested to record this interview, if possible with video and audio, in order to transfer it to Mr. Luis Pablo Yo Secaida, so that he can carry out the corresponding analysis. If you agree with this, I will turn on the recorder right now.*(Interviewer starts recording)* . *(Continuous reading, which must be recorded on video and audio / audio)* By agreeing to answer this interview, you authorize Mr. Luis Pablo Yon Secaida to use the answers and information for the preparation of his postgraduate thesis, as well as inputs for the formulation of a methodological guide for other interviews and surveys. At no time during the other interviews and surveys will your personal information or responses be revealed. However, your name may be mentioned in the thesis to be carried out. At the end of the thesis, a copy of it will be sent to you. Would you agree?

---

**interview Ballot**

---

| Section 1 | Personal data and housing |
|---|---|

*To the interviewer:*

*In this part of the interview, the main objective is to know a little about his life and the space in which he grew up. We want to understand how it changed, and if this influenced their cureent life. It is extremely necessary that the subject draws the spaces where he lived, as well as the space outside and around. Could it be that the activity of the black salt influenced his day-to-day development? If possible, it would be good if he drew a **sketch** , or, description of the entire space. Each of the houses in which he has lived must be described, along with a sketch if possible*

1 **Where exactly were you born?**
*(Department, municipality, name of the community)*
2 **How many people reside in your household?**
*(Enter amount)*
3 **Who resides in your household?**
*(Write down the roles, understand marital status, children, grandparents, in-laws, grandchil*
4 **What do you do?**
*(Ask for each member mentioned above)*
5 **Have you lived in this house all your life?**
*(If you answer yes, skip to question 7)*
6 **Can you describe the homes where you have previously lived?**

*(Write down the amount of dwelling and its characteristics: number of rooms and what they were (example 1 kitchen, 1 bedroom, a bathroom outside the dwelling, a patio).*
7 **What are the things you remember from that stage of your life?**
*(This applies to events that marked your childhood WITHIN the territory of your home)*
8 **Why did you decide to move out of the house?**
*(Ask for each of the dwellings described in questions 5 and 6)*
9 **Describe your current dwelling?**
*(Here you can make a drawing (sketch) of the house. You must describe, or draw a sketch of each house that you have lived in your life)*

| Section 2A | Black Salt Industry |
|---|---|

*To the interviewer: This section only applies to current black salt workers.*
*In this part of the interview, the main objective is to learn as much as possible about black salt. Information related*
*not so much about its realization (creation process), but about the space where it is created, the people involved, how*
*the people perceive this product, etc… It is necessary to know exactly where the salt beaches are, and where they were*
*(it will be given a general map so that the subject can mark the locations of the beaches)*

**10 How was black salt introduced?**

**11 What role did your family have in teaching you the process of creating black salt?**

**12 How many people previously made black salt?**

**13  Was it possible to live only on black salt? Because otherwise?**

**14 How has its price/value changed?**

**15 Previously, how was the interest of people towards black salt?**

**16 In your opinion, what had the biggest impact on the black salt industry?**
   *(Important question, we want to know what influenced the loss of culture)*
**17 This graphic shows several events that were documented about black salt, is it true?**
   **Could you expand it? Is information missing?**
   *(The graph is attached to this document)*
**18 Are there other influences NOT investigated, that influenced the loss of culture?**

**19 Has this space used (for the creation of black salt) changed over time?**

**20  Are you the owner of said space?**
   *(Mark your exact location on a map)*
**21 How was it before, how is it now?**

**22  Is there a risk of losing the beach where you make the black salt?**
   (If you answer no, skip to question 25).
**23 Is this risk constant?**
   (If you answer no, skip to question 25).
**24 If it were lost, what would be the consequences?**

**25 How do you see the future of black salt?**

| Section 2B | Black Salt Industry |
|---|---|

*For the interviewer: This section only applies to* **former workers** *of black salt.*
*In this part of the interview, the main objective is to learn as much as possible about black salt. Information*
*not so much about its realization (creation process), but about the space where it is created, the people*
*involved, how the people perceive this product, etc… It is necessary to know exactly where the salt beaches*
*are, and where they were (it will be given a general map so that the subject can mark the locations of the*
*beaches)*

**10 How was black salt introduced?**

**11 What role did your family have in teaching you the process of creating black salt?**

**12 How many people previously made black salt?**

**13  Was it possible to live only on black salt? Because otherwise?**

**14 How has its price/value changed?**

**15 Previously, how was the interest of people towards black salt?**

**16 In your opinion, what had the biggest impact on the black salt industry?**
   *(Important question, we want to know what influenced the loss of culture)*
**17 This graphic shows several events that were documented about black salt, is it true?**
   **Could you expand it? Is information missing?**
   *(The graph is attached to this document)*
**18 Are there other influences NOT investigated, that influenced the loss of culture?**

**19 Are you or were you the owner of a salt beach?**

   *(Mark your exact location on a map)*
**20 How did your life change when you left the black salt industry?**

**21 Was it financially beneficial to leave the black salt industry?**

**22 Have your daily activities changed in any way?**

**23 Do you miss working in the black salt?**

**24 How do you see the future of black salt?**

| Section 3 | People related to the industry |
|-----------|-------------------------------|

*In this part of the investigation, the objective is to be able to find more information about people who **currently is making** the black salt. Documentation found online mentions only Don Max, however photos and small comments describe other people involved. It is also necessary to know the point of view of the people who were involved in black salt in the past. What is the reality of these people? How did your life change due to the change of profession?*

25 **Are you aware of other people who make black salt?**
(Inquire if it is possible to find these people and thus carry out this survey)

26 **Are you aware of other people who MADE black salt?**
(Inquire if it is possible to find these people and thus carry out this survey)

27 **Are these people still in Sacapulas or did they move out of town?**

28 **Are your salt beaches still usable, or is it too late?**

29 **Can we contact these people in any way?**

| Section 4 | Black Salt Beaches |
|-----------|-------------------|

*In this part of the interview, we want to know everything that exists **around** of the salt beaches. Due to the loss of these spaces for various reasons (questions from the previous section), we want to know what the new use of these spaces is, or if they are just idle. It is important to catalog the type of activity around these spaces. Are they used by children as a play area? Are they used for x,y,z activity?*

30 **Can you remember the black salt beaches that were used in the past?**
(Mark the location of the salt beaches on the map)

31 **Of the salt beaches that you point out, which ones are no longer used?**
(Mark the location of the salt beaches on the map)

32 **How are these spaces used?**
(Ask specifically if the town has reclaimed these spaces in the form of informal use, or if they are simply idle spaces)

33 **Around the black salt beaches, do the owners live?**

34 **The spaces around of the black salt beaches were used for its production?**
(Ask if the industry or other activity was developed in the surroundings of the beaches)

35 **The spaces around of the salt beaches, are they used today?**

| Section 5 | Religion and the Rio Negro |
|-----------|---------------------------|

*In this part of the interview, what we are looking for specifically is the **Relationship between Mayan religion, ancient product and current reality**. Due to the introduction of the Christian religion, the Mayan religion had to be adapted in one way or another. Today, documentation presents information about a certain religious value to the river. However, this is scarce. If possible, this aspect should be investigated deeply.*

36 **Does the Rio Negro and the elaboration of salt have any religious significance?**
(Ask about a specifically Mayan religious value, syncretism.)

37 **Documents (report from the 70s) mention 2 nahuales that protect the river. Are you aware of**
(Find your exact description of the Mayan mythology around the lake. Extend if necessary)

38 **Are there annual celebrations in relation to the river?**
(If YES, go to question 39. If NO, go to question 45.)

39 **Please describe in as much detail as possible the celebration in question.**

40 •**Has this celebration changed in size and in people who celebrate it?**

41 **What religion professes this celebration?**

42 **Do people have any attachment to these religious festivals?**

43 **Is the whole town part of these festivities?**

44 **What do you think is the opinion of the people towards this celebration?**

45 **Is there any other type of celebration related to black salt?**

**Appendix C. An Example of the Microanalysis Tool Used in the Interview, Where the Time, Interviewee and Concept Are Organized**

| Concept | < Terrain > | |
|---|---|---|
| Definition | Information | |
| **Iterations** | PSS-A-1,2 | 00:28<br>I: At what point was all that space filled?<br>A-1: When they came, they left the area...; the bosses didn't come anymore... to plant or something like that. And others left the place. |
| | | 24:18<br>I: Do you remember more or less where the salt was before?<br>A-1: Yes, there on the beach, where that hole was (area C). There were salt pans belonging to an owner that was there. And that's where it started, where there was a mangal. |
| | PSS-D | 00:07<br>D: The earth stayed and go, who is going to activate<br>That's why we needed to study more, we needed to work more. |
| **Theoretical notes** | | Those involved do not have exact knowledge of where each person's land is located.<br>No specific dates were mentioned, so there is a gap at this time. |

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
