# Peer review of "Current Situation of Traditional Architecture Located inside Cultural Mayan Heritage Spaces in Remote Villages of Guatemala: Case of the Black Salt Kitchens"

_sustainability, doi:10.3390/su16083194_

Round 1
Reviewer 1 Report
Comments and Suggestions for Authors
A big part of the paper repeats the information from the paper of the same authors: Yon Secaida LP, Mori S, Nomura R. - Assessment of Natural Disasters Impact on Cultural Mayan Heritage Spaces in Remotes 1623 Villages of Guatemala: Case of Black Salt. Sustainability. 2023; 15(16):12591. https://doi.org/10.3390/su151612591.
Even the text was somehow changed, too many figures, graphics, material are identical (more than 20% from the total). I consider those parts have to be reduce or replaced with original information and to indicate the previous publication (being the same research, splitted in different parts).
For the rest of issues, I have marked directly in the .pdf file. Are some technical suggestions and is necessary the widening of the perspective. It is mandatory to have a holistic approach, respectively what benefits could bring to keep them "in life" or to spend money for this, not for the roads, health, safety or measures against flooding.
Maybe a small project for a video documenting of the procedure and a virtual 3D tour in the nearest museum could be a cheaper and faster solution. This will be able to provide a valuable reference for the future, if it is desired to build a physical salt kitchen.

Comments on the Quality of English LanguageAuthor Response
Thanks for the comments, i uploaded a pdf where you can see the changes and responses on my behalf.
Thanks for the interest and help.

Reviewer 2 Report
Comments and Suggestions for Authors
The paper describes extensive field research in a highly sensitive geographical area.
It presents the on-desk method and study and field work. The results are consequential to the research activities. The field approach is interesting with a specific survey and the complete survey of the Salt Kitchen.
Comments on the Quality of English LanguageI can suggest an in-depth revision of the English Language, mainly in chapters 1 and 2, because some sentences are cut and sometimes with a narrative so difficult to read.
Author Response
Thanks for the comments, i reviewed the english of the sections mentioned. I also expanded certain parts of the paper by the comments of the other reviewers.
Thanks for the support.
Reviewer 3 Report
Comments and Suggestions for Authors
Based on the current situation of traditional architecture in the Mayan cultural heritage space of remote villages in Guatemala, this article proposes a survey on the layout and structural forms of traditional architecture of black salt, as well as an analysis and research on the history and current situation of black salt. The aim is to explore, protect, and inherit the local Mayan historical and cultural heritage in remote areas. The black salt kitchen direction of this study has certain innovation and academic research value, and has a certain value in the protection and utilization of historical architectural heritage. However, further improvements are needed in some details.
1. It is recommended to supplement the various locations of other black salt production areas in Guatemala and conduct horizontal comparisons to highlight the unique value of selecting and researching this black salt production area.
2. It is recommended to provide a detailed introduction to the research methods, such as “between-method triangulation”.
3. It is recommended to organize the abstract section, which can include the research process and methods, and highlight the summary content of the key points of the entire text.
4. It is recommended to insert some pictures in the “Features of the salt kitchen” and “Architecture features of the salt kitchen” sections, with illustrations and text to facilitate readers' understanding of the structural features.
5. It is recommended that “Methodology diagram” in Figure 5 be appropriately concise and clear. Various connections and multi paragraph text expressions are not conducive to reading comprehension.
6. It is recommended to arrange the original photos of the shooting site into a group image for readers to read and layout beautifully.
7. In the conclusion section, it is suggested to discuss and supplement the specific research directions and plans for the next step. For example, since the decline of the black salt industry is caused by the era and has irreversibility, should we consider the timeliness after the revival? How to balance the economic feasibility of reconstruction with the increase of local floods and other natural disasters? How to preserve the transformation and upgrading of structure or function that needs to be made? Is there still thinking about the addition of modern salt production methods, rather than just the outdated production methods of building modern houses? and so on.
8. It is recommended that the next step of the plan is to try to verify the accuracy of the inference of the " Relationship between architectural elements and space" and the " The hypothesis of the original black salt beach ".
Author Response
Thank you for your comments, i have attached a PDF file with the explanations of the changes made in the manuscript.
Thanks for the time and interest

Round 2
Reviewer 1 Report
Comments and Suggestions for Authors
Dear authors, I want to congratulate you for your hard work in revising and completing the manuscript. Almost every suggestion or observation was carefully revised, valuable information and enlightening pictures were included, and the quality of the article have increased considerably.
In addition, now it is much easier to understand your contribution to this topic and the importance of the approach to preserve at least the existing information and data about the topic.
The conclusions have been correctly re-formulated and these highlight the obtained results, and also the non-covered aspects of the research. Most suitable recommendations were included and also the perspective for the subjected area.
Thank you for your professional response to the comments and suggestion during the revision phase and for the neutral tone. Also thank you for the chance to know more about this subject and to contribute to the increasing of the quality of your paper.
(Are only few (re)marks or typing issues in the attached file - rows 110, 131, 158, 439, 459, 477, 649, 885-886)/
